# Enhanced pericyte-endothelial interactions through NO-boosted extracellular vesicles drive revascularization in a mouse model of ischemic injury

Ling Guo [1,2,6] ✉, Qiang Yang[1,6], Runxiu Wei[1], Wenjun Zhang[1], Na Yin[1], Yuling Chen[1], Chao Xu[2], Changrui Li[3], Randy P. Carney [4] ✉, Yuanpei Li [5] ✉ & Min Feng [1] ✉

Despite improvements in medical and surgical therapies, a significant portion of patients with critical limb ischemia (CLI) are considered as "no option" for revascularization. In this work, a nitric oxide (NO)-boosted and activated nanovesicle regeneration kit (n-BANK) is constructed by decorating stem cell-derived nanoscale extracellular vesicles with NO nanocages. Our results demonstrate that n-BANKs could store NO in endothelial cells for subsequent release upon pericyte recruitment for CLI revascularization. Notably, n-BANKs enable endothelial cells to trigger eNOS activation and form tube-like structures. Subsequently, eNOS-derived NO robustly recruits pericytes to invest nascent endothelial cell tubes, giving rise to mature blood vessels. Consequently, n-BANKs confer complete revascularization in female mice following CLI, and thereby achieve limb preservation and restore the motor function. In light of n-BANK evoking pericyte-endothelial interactions to create functional vascular networks, it features promising therapeutic potential in revascularization to reduce CLI-related amputations, which potentially impact regeneration medicine.

Critical limb ischemia (CLI) is the advanced stage of peripheral artery disease whereby blockages within the arteries restrict the flow of blood to the lower extremity, which is associated with high rates of amputation and mortality[1]. Diabetes, hypertension, and dyslipidemia raise the risk of developing CLI. Given the anticipated rise in the prevalence of these diseases, CLI is becoming an increasingly important issue. Patients with CLI have been reported to have a major amputation rate up to 40% at 6 months and a mortality rate of approximately 25% in the first year after presentation[2]. The high amputation rates in the CLI

patients result in a poor quality of life and a large overall healthcare burden[3]. Surgical and catheter-based revascularization to restore perfusion of the extremity is the fundamental therapies for limb preservation in patients with CLI[4]. However, approximately 30% of these patients are not indicated for any intervention due to heavily vascular calcifications, severe cardiovascular comorbidities, or other complicating pathological conditions[1,5]. This results in a significant portion of CLI patients considered as "no option" for revascularization, with no other alternative therapy capable of reducing the need for

[1]School of Pharmaceutical Sciences, Sun Yat-sen University, University Town, Guangzhou 510006, P.R. China. [2]Key Laboratory of Tropical Biological Resources of Ministry of Education, School of Pharmaceutical Sciences, Hainan University, Haikou 570228, P. R. China. [3]Guangzhou Zhixin High School, Zhixin South Road, Guangzhou 510080, P.R. China. [4]Department of Biomedical Engineering, University of California Davis, Davis, CA 95616, USA. [5]Department of Biochemistry and Molecular Medicine, UC Davis Comprehensive Cancer Center, University of California Davis, Davis, CA 95616, USA. [6]These authors contributed equally: Ling Guo, Qiang Yang. ✉e-mail: guoling@hainanu.edu.cn; rcarney@ucdavis.edu; lypli@ucdavis.edu; fengmin@mail.sysu.edu.cn

amputation[6]. Hence, there is an urgent unmet medical need for the development of therapies for CLI to avoid amputations and preserve limb function.

Newly formed vessels are composed of two primary cell types, endothelial cells and pericytes. Endothelial cells form the inner lining of the vessel wall and pericytes envelop the surface of the vascular tube. Revascularization by establishing functional and stable collateral vessels that augment ischemic tissue perfusion is a complex process mainly involving proliferation and migration of endothelial cells, and pericyte recruitment to form functional collateral vessels[7]. At the present time, therapeutic angiogenesis for revascularization has been only slightly successful[8]. One reason for the disappointing clinical results is that the efforts typically focus on administration of single proangiogenic growth factors or their genes, such as vascular endothelial growth factor (VEGF), fibroblast growth factor (FGF) and hepatocyte growth factor (HGF), to activate endothelial cells. Yet, these approaches are not sufficient to recapitulate the complex angiogenesis processes arising from cooperative cell interactions during the establishment of a functional collateral network[9,10]. Additionally, although stem cell therapy has shown potential for repair of ischemically damaged tissues, clinical therapies with stem cells face considerable challenges with regard to poor cell survival in the ischemic tissues and the possibility of teratoma formation[11]. On the other hand, nanoscale membrane-bound extracellular vesicles (EVs) derived from mesenchymal stem cells (MSCs) traffic a complex array of angiogenic factors and have been shown to recapitulate the therapeutic effects of MSCs without concerns about oncogenesis[12–14]. Furthermore, we recently demonstrated that secreted EVs inherited SNARE proteins from parent cells acting as a driving force for membrane fusion, resulting in rapid delivery of therapeutic cargo into cells[15]. Thereby, we hypothesize that MSC-derived EVs carrying multiple angiogenic factors provide greater accessibility to endothelial cells for revascularization.

Besides endothelial cells, pericytes are the ensheathing cells that wrap around capillary walls and have emerged as key regulators of angiogenesis. Owing to their contractile capabilities, pericytes have mainly been associated with stabilization and hemodynamic processes of blood vessels[16,17]. A growing body of evidence suggests that enhancing the maturity of newly formed vessels by active participation of pericytes is essential to the clinical success of revascularization[18,19]. Pericyte establishment ensures formation of leakage-resistant blood vessels, which can support blood flow. Methods to recruit pericytes to newly formed vessels are of great interest to realize this robust therapeutic approach. The signaling molecule nitric oxide (NO) is able to efficiently induce pericyte recruitment as well as subsequent morphogenesis and stabilization of immature endothelial vessels[20]. However, NO synthesis is significantly impaired in peripheral arterial diseases[21]. It has been documented that NO in the vasculature is synthesized by endothelial nitric oxide synthase (eNOS), and NO effects in angiogenesis are primarily mediated via activation of eNOS[22]. Glyceryl trinitrate (GTN) acts as an eNOS agonist that enables to activate eNOS to produce NO in endothelial cells[23,24]. It also serves in its well-known capacity as a vasodilator to improve blood flow to ischemic areas of the myocardium[25]. So far, the application of NO in therapeutic angiogenesis has yet to be explored in depth[26,27]. Thus, given the effects of eNOS activation and vasodilation, we explore combining GTN with MSC-derived EVs as a NO booster to enhance regeneration of well-functioning collateral vessels in the severe ischemic limbs by pericyte recruitment and an increase in blood flow.

Here, we constructed a nitric oxide (NO)-boosted and activated nanovesicle regeneration kit (n-BANK) by decorating MSC-derived EVs with NO-nanocages for efficient revascularization to address the problem of amputation following CLI. n-BANKs shuttling NO-nanocages and multiple angiogenic factors augmented eNOS-derived NO synthesis and stored NO in endothelial cells. Subsequently, eNOS-derived NO was withdrawn to robustly recruit pericytes for the stabilization of nascent endothelial cell tubes, giving rise to mature blood vessels that exhibited desirable functions in contractility and dilatation. Notably, n-BANKs energized pericyte-endothelial interactions via NO to induce sustained dilation of blood vessels for supplying the bulk of flow to the ischemic tissues, while attenuation of these interactions led to vessel dysfunction. As proof of principle, we evaluated whether the n-BANKs conferred revascularization effectively in an animal model of CLI. Our results demonstrated that n-BANKs completely normalized blood flow in the ischemic area following CLI, and thereby achieved limb preservation and restored motor function.

## Results

CLI, in which blood flow is not sufficient to maintain tissue viability, is the leading cause of nontraumatic amputation[28]. There is clinical evidence to suggest that establishment of functional and stable collateral vessels composed of endothelial cells and pericytes can help to supply sufficient blood flow to ischemic territories[29]. Our evidence indicated that vascular endothelial cells were impaired with a loss of eNOS activity, and pericyte dropout from the microvessels presented within the ischemic tissues in a mouse model of severe hindlimb ischemia (Fig. 1A, B and Supplementary Fig. 1). To develop collateral vessels after ischemic injury, we chose the EVs derived from activated MSCs with the highly expressive genes of several pro-angiogenic growth factors by resveratrol stimulation and separated by sequential centrifugation, as well as overcoming the major limitations (low survival rates) of transplanted MSCs (Fig. 1C, D and Supplementary Figs. 2 and 3)[30]. The activated EVs rich in pro-angiogenic growth factors such as VEGF exerted the pro-survival effects in injured endothelial cells which could assemble into capillary-like tubes (Fig. 1E, F and Supplementary Fig. 4). On the other hand, pericyte recruitment during vasculogenic tube assembly is an essential step in establishing functional collateral vessels[18]. Motivated by the fact that eNOS-derived NO exerts direct effects in pericyte recruitment[31], we measured the eNOS-derived NO's capacity to regulate pericyte recruitment by using GTN, an eNOS agonist that enable to stimulate eNOS activity of endothelial cells in NO production (Fig. 1G)[32]. eNOS-activated endothelial cells showed more effective pericyte recruitment than ischemia-injured endothelial cells (Fig. 1H).

### Construction of n-BANKs by decorating MSC-derived EVs with NO-nanocages

We reasoned that rescuing damaged endothelial cells and augmenting eNOS activity for pericyte recruitment could be exploited to effectively regenerate functional collateral vessels for CLI treatment. To this end we constructed n-BANK, a NO-boosted and activated nanovesicle regeneration kit, by modifying activated MSC-derived EVs with NO-nanocages to enhance regeneration of well-functioning collateral vessels for CLI revascularization (Fig. 1I). Firstly, the NO-nanocages were prepared by encasing GTN, an eNOS activator, in albumin nanocages. GTN trafficking into endothelial cells by albumin nanocages was more efficient than GTN alone (Fig. 1J), mostly likely due to that albumin could bind vascular endothelial cells with high affinity[33]. By using molecular docking analysis and surface plasmon resonance (SPR) binding analysis, we revealed that GTN had a good binding affinity to albumin, and packaging of GTN into albumin nanocages was dependent on hydrogen bonding and electrostatic interactions, as shown in Fig. 1K and Supplementary Fig. 5A. The albumin molecule, with a variety of amino acid residues, had four binding sites for GTN molecules. It is likely that these binding sites assist in the self-assembly process of the GTN-loaded nanocages. Further analysis showed that the uniform-sized NO-nanocages of $8.66 \pm 0.13$ nm in diameter, with a spherical shape were slightly larger than albumin control (Fig. 1L, M).

Next, activated MSC-derived EVs were harvested by sequential centrifugation as previously described with some modifications[34]. The

TEM images and NTA analysis showed the isolated activated EVs had cup-like concavity structure with an average diameter of 152.87 nm. To drive coupling of NO-nanocages to activated MSC-derived EVs for n-BANK formation, phosphatidylcholine (PC), having a great affinity to bind to albumin of NO-nanocages by hydrogen binding, was inserted into the EV membrane (Fig. 2A). We confirmed that PC indeed showed relatively strong binding to albumin of NO-nanocages by surface plasmon resonance binding analysis (Fig. 2B). The n-BANKs with 28.2% NO-nanocages tethered to the surface of EVs were prepared by using a co-culture method (Supplementary Fig. 5B) and monitored via changes

of the intrinsic fluorescence spectrum of albumin with an emission maximum at 341 nm, which led to a loss of fluorescence signal, compared with NO-nanocages (Fig. 2C). The loss of fluorescence signal was likely caused by the cation-π interactions between the choline ammonium groups of phosphatidylcholines in the membrane of EVs and the tryptophan residues in albumin of NO-nanocages, resulting in quenching of the intrinsic fluorescence of tryptophan residues[35]. By using confocal fluorescence microscopy, we observed the colocalization of NO-nanocages (green) and EVs (red) while tracking the n-BANKs, as shown in Fig. 2D. Subsequent TEM images and NTA analysis

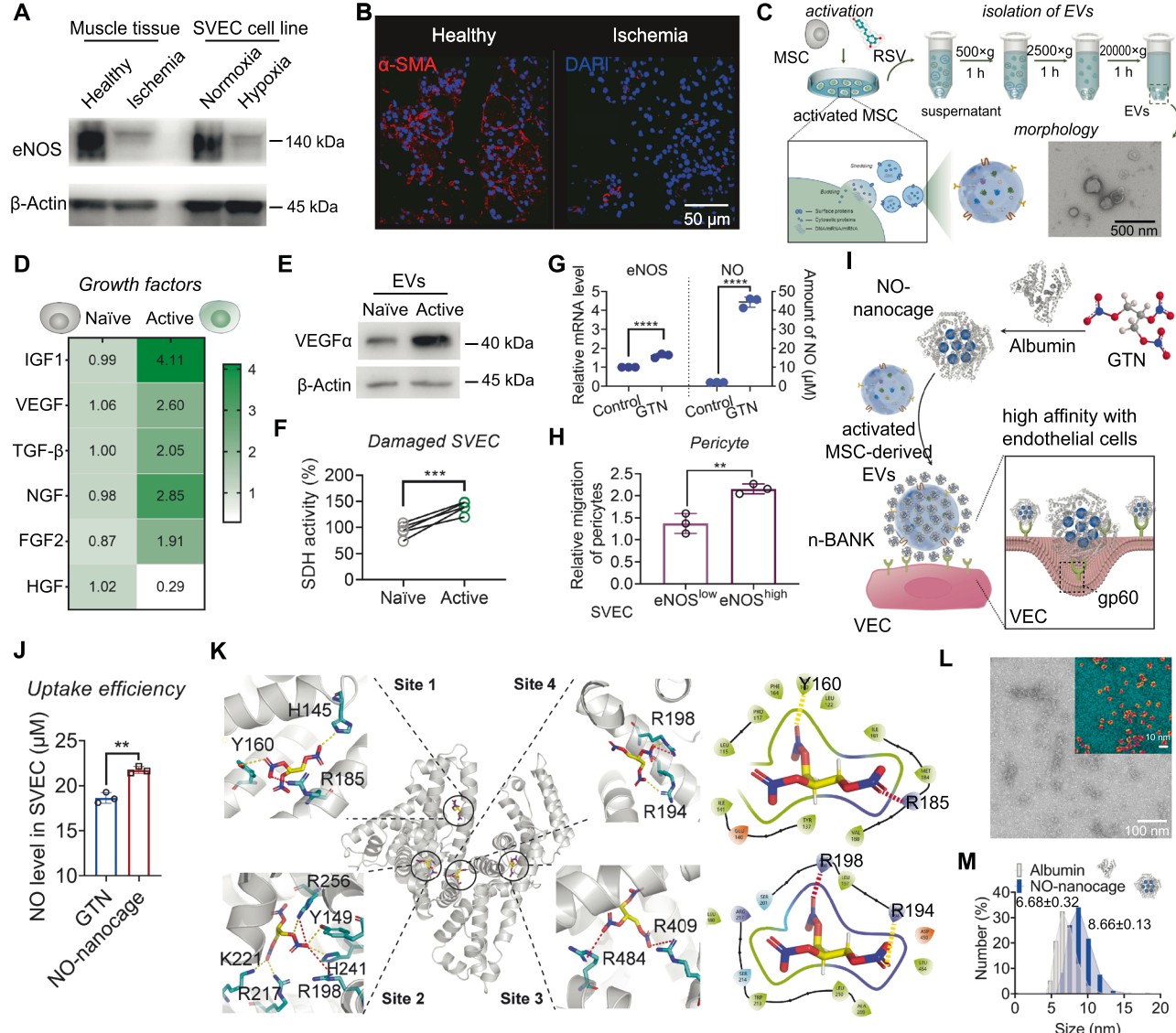

**Fig. 1 | Construction of n-BANKs by decorating MSC-derived EVs with NO-nanocages. A** Western blot of eNOS and β-actin with protein lysates from muscle tissues and SVEC cells. SVEC4–10 cells were incubated in the present of 500 µM cobalt chloride (CoCl₂), to mimic ischemic conditions ($n = 3$). **B** Representative immunofluorescence images of muscle sections stained with α-SMA (red) and Dapi (blue) ($n = 3$). Scale bar, 50 µm. Muscles were extracted from healthy or ischemic hindlimb sections of mice at day 3 after surgery. **C** Illustration of isolation of extracellular vesicles from the supernatants of activated MSCs by sequential centrifugation. Morphology of extracellular vesicles were detected by transmission electron microscope (TEM) ($n = 3$). Scale bar, 500 nm. **D** Heat map showing the expression patterns of angiogenic growth factor genes in naïve and active MSCs. **E** Western blot of VEGFα and β-actin with protein lysates from naïve and active MSCs ($n = 3$). **F** SDH activity of damaged SVEC cells treated with EVs isolated from

naïve and active MSCs ($n = 5$; **$P = 0.0004$). **G** Relative quantification of *eNOS* expression in SVEC cells using qRT-PCR and release of nitric oxide in cell supernatant of SVEC cells quantified by Griess reagent ($n = 3$; ****$P < 0.0001$). **H** Migration of pericytes quantified by real-time cell migration assays ($n = 3$; **$P = 0.0058$). **I** Schematic illustration of the construction of n-BANK. **J** NO levels in SVEC cells treated GTN or NO-nanocage by Griess reagent ($n = 3$; **$P = 0.0016$). **K** Detail interactions between albumin and GTN by molecular docking analysis. Site 1, 2, 3, and 4 represented four binding sites between albumin and GTN involved in hydrogen bonding (yellow dotted lines) and electrostatic interactions (red dotted lines). **L** TEM imaging of NO-nanocages (pseudo-colored, NO-nanocages: red, substrate bottom surface: cyan; $n = 3$). Scale bar, 100 nm. **M** Size distribution of albumin and NO-nanocages. Data are mean ± SD, ** is $P < 0.01$, *** is $P < 0.001$, **** is $P < 0.0001$ by two-tailed Student's $t$ test.

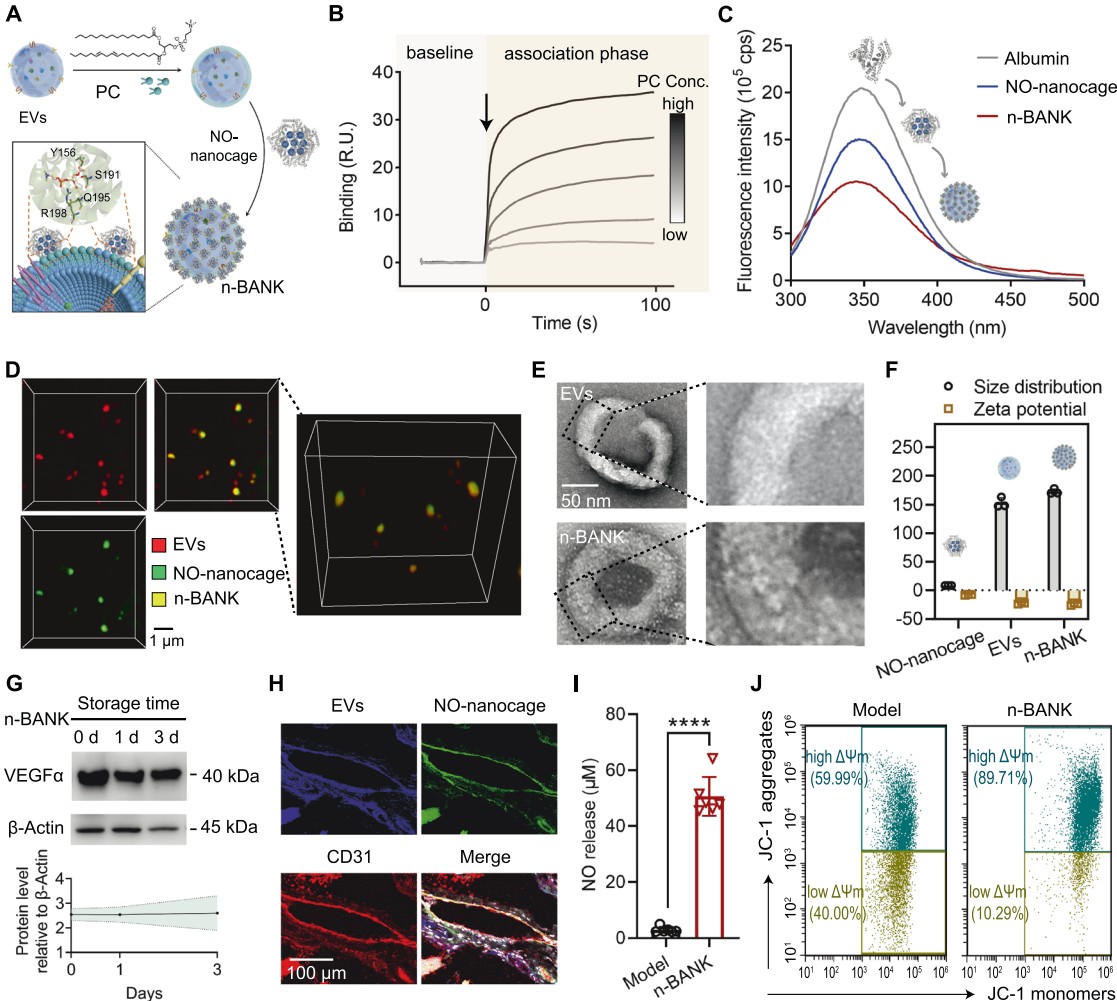

**Fig. 2 | Characterization of n-BANK and its effects on damaged endothelial cells. A** Schematic illustration of the construction of n-BANK. **B** Surface plasmon resonance measuring the binding capacity of albumin and PC. **C** Changes of fluorescence intensity among albumin, NO-nanocages, and n-BANKs. **D** Fluorescent analysis of n-BANKs by laser scanning confocal microscope ($n = 3$). EVs were stained with DiI (red) and NO-nanocages were labeled with FITC (green). Scale bar, 1 μm. **E** TEM images of EVs and n-BANKs ($n = 3$). Scale bar, 50 nm. **F** Size distribution and zeta potential of NO-nanocages, EVs, and n-BANKs ($n = 3$). **G** Stability of VEGFα protein in n-BANKs during storage was assessed by western blot analysis ($n = 3$). **H** Representative immunofluorescence images of ischemic muscle tissue from n-BANK treated mice ($n = 3$). The n-BANKs were constructed with FITC-conjugated NO-nanocages (Green) and DiD-labeled EVs (blue), and the ischemic muscle sections were stained with CD31 (Red). Scale bar, 100 μm. **I** Release of nitric oxide in cell supernatant of SVEC cells using Griess reagent ($n = 6$; **** is $P < 0.0001$ by two-tailed Student's $t$ test). **J** Mitochondrial membrane potential (ΔΨM) in SVEC cells in the present of 500 μM CoCl$_2$ by flow cytometry using the JC-1 mitochondrial probe. Data are mean ± SD.

revealed the detailed morphology of n-BANKs was EVs surrounded by NO-nanocages (Fig. 2E), and the average particle size of n-BANKs was measured to be 172.87 ± 4.71 nm (Fig. 2F). The large 20-nm increment in size of n-BANKs compared to unmodified EVs was attributable to surface attachment of NO-nanocages. The content of VEGFα, a critical growth factor in angiogenesis, in n-BANKs remained stable during the storage period (Fig. 2G).

To evaluate the biodistribution and in vivo stability of n-BANKs in ischemic tissues, the dual fluorescence-labeled n-BANKs, composed of FITC-labeled NO-nanocages and DiD-labeled EVs, were administered (i.m.) into BALB/c mice with severe ischemic hindlimbs. The real-time in vivo imaging system demonstrated that n-BANKs stayed at the ischemic hindlimb for nearly 60 h (Supplementary Fig. 6). Then, immunofluorescence staining of frozen sections for the endothelial cell marker CD31 was obtained at 24 h after administration. CD31 staining was observed primarily at the margins of the injured blood vessel. Strikingly, n-BANKs were found to accumulate in the CD31-positive endothelium (Fig. 2H), most likely because the albumin presented on n-BANKs could bind to albumin-binding glycoprotein

(gp60) expressed specifically on the surface of vascular endothelial cells[33]. Meanwhile, the abundant endogenous albumin could limit unwanted interactions between NO-nanocages of n-BANKs and endogenous molecules. Furthermore, n-BANKs markedly elevated NO production and the positive mitochondrial membrane potential (ΔΨM) of ischemically damaged endothelial cells (Fig. 2I, J and Supplementary Fig. 7A, B), which are involved in vascular regeneration from ischemic injury. Taken together, these data suggested that n-BANKs were constructed by EVs decorated with NO-nanocages, which could attach to the damaged endothelium and rescue endothelial cells from ischemia-induced damage.

## n-BANKs activated endothelial cells and promoted endothelial tube formation

The first step in angiogenesis occurs by the formation of a new sprout off of the preexisting vessels, mediated by endothelial cell migration[36]. Thus, we sought to assess whether n-BANKs could enhance endothelial cell migration. The scratch assay was performed to study vascular endothelial cell migration in vitro, which mimicked to some extent

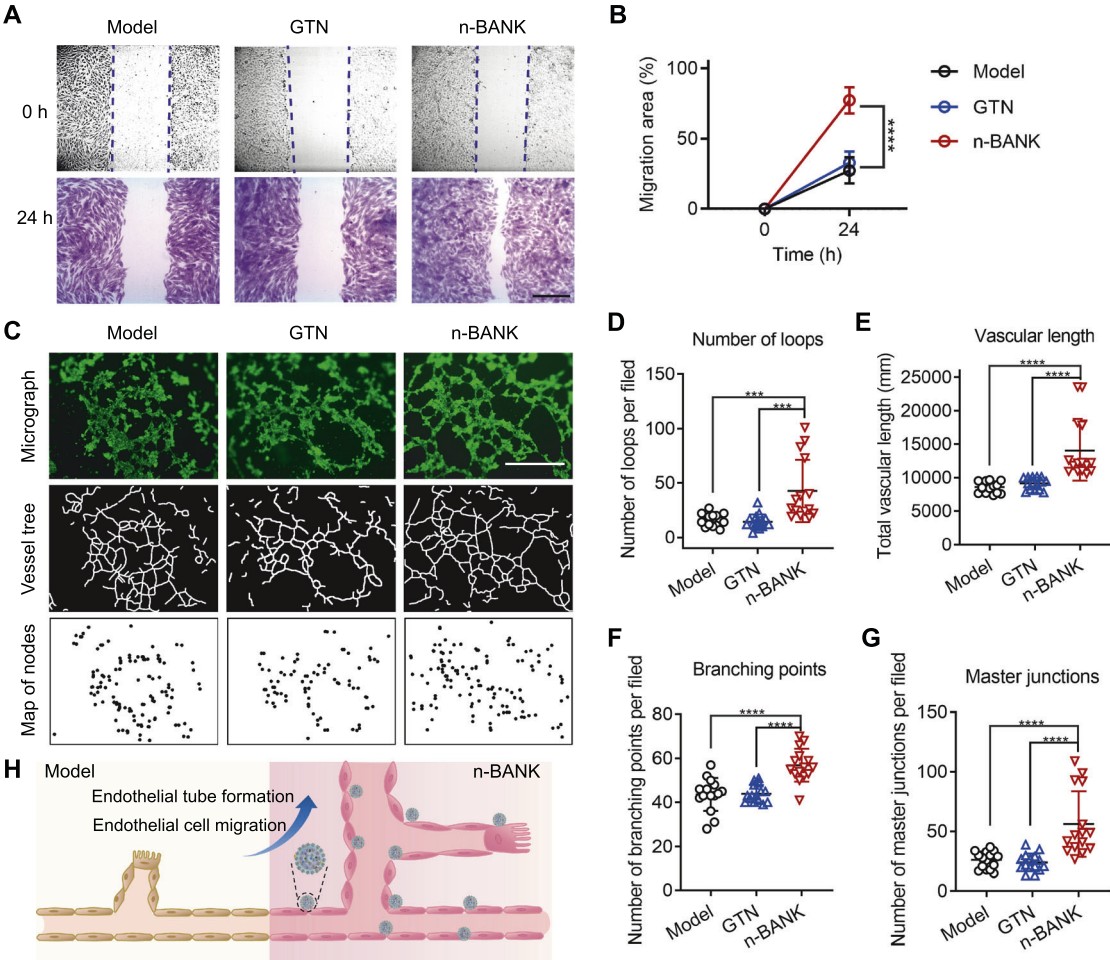

**Fig. 3 | n-BANKs promoted endothelial tube formation. A** Representative photomicrographs from a scratch-cell motility assay of SVEC4−10 cells in the present of 500 μM $CoCl_2$ for the indicated times ($n = 8$). The scratch area reflected the endothelial cell migration ability. Scale bar, 400 μm. **B** Quantification of SVEC4−10 cell migration rate in a scratch wound assay ($n = 8$; ****$P < 0.0001$). **C** Tube formation was monitored periodically with images depicting endothelial tube formation 4 h after seeding ($n = 15$). The number and length of tubules assembled by elongation and joining of endothelial cells reflected the tubulogenesis ability of n-BANKs. Scale bar, 1000 μm. **D** Quantification of number of loops at 4 h ($n = 15$; Model, ***$P = 0.0004$; GTN, ***$P = 0.0002$). **E–G** Quantification of vascular progression (**E** vascular length; **F** number of branching points; **G** number of master junctions) at 4 h ($n = 15$; ****$P < 0.0001$). **H** Schematic illustration of endothelial tube formation activated by n-BANKs. Data are mean ± SD, *** is $P < 0.001$, **** is $P < 0.0001$ by one-way ANOVA test.

migration of cells during angiogenesis in vivo. As shown in Fig. 3A, B, endothelial cells stimulated by n-BANKs migrated significantly farther than cells exposed to medium alone after 24 h in conditions that mimic ischemia ($P < 0.0001$). However, GTN and NO-nanocages had no impact on the endothelial cell migration (Fig. 3A and Supplementary Fig. 7C). The promigratory effects of n-BANKs were further confirmed under normoxia (Supplementary Fig. 8). Having established that n-BANK stimulated endothelial cell migration, we next evaluated the ability of n-BANKs to induce endothelial tube formation using in vitro tube formation assay. SVEC4−10 vascular endothelial cells were grown in Matrigel and stained with calcein acetoxymethyl ester (Calcein-AM) to visualize tube formation. The endothelial cells incubated in medium tended to cluster together, forming large aggregates in the absence of tubular organization, which is consistent with previous studies[37]. Strikingly, n-BANKs induced marked tubulogenesis, with the formation of tubules assembled by elongation and joining of endothelial cells, compared with either the GTN treated or untreated control group (Fig. 3C). Quantitative analysis of the data further confirmed that n-BANKs substantially enhanced tubular formation of endothelial cells, resulting in increases of 263.4% in number of loops ($P < 0.001$), 163.3% in vascular length ($P < 0.001$), 130.4% in number of branching points ($P < 0.001$), and 215.1% in number of master junctions ($P < 0.001$), in

comparison with untreated endothelial cells (Fig. 3D–G). Collectively, these data suggested that n-BANKs could activate endothelial cells and subsequently promote endothelial tube formation (Fig. 3H).

## n-BANKs augmented eNOS activity of endothelial cells to facilitate pericyte recruitment

Mature microvascular structure is characterized by pericytes wrap around the endothelial tubes to provide structural support and regulate vascular tone. Evidence shows endothelial-derived NO can induce pericyte recruitment to endothelial tubes[20,38]. Thus, we sought to investigate whether n-BANKs could activate eNOS by promoting its phosphorylation for production of endothelial-derived NO to recruit pericytes (Fig. 4A). As seen in Fig. 4B, the addition of n-BANKs markedly stimulated *eNOS* mRNA expression (by 2.27 times) in damaged vascular endothelial cells ($P < 0.0001$). And the stimulatory effects of n-BANKs were enhanced over time (Supplementary Fig. 9A). Notably, overexpression of eNOS protein at 4 h, accompanied with a marked increase in eNOS phosphorylation, occurred at as early as 10 min in n-BANK treated SVEC4−10 cells (Fig. 4C and Supplementary Fig. 9B). Next, we detected endothelial-derived NO production in damaged SVEC4−10 cells. The results demonstrated that the NO-producing endothelial cells had a 3.13-fold increase in 4 h after treatment with

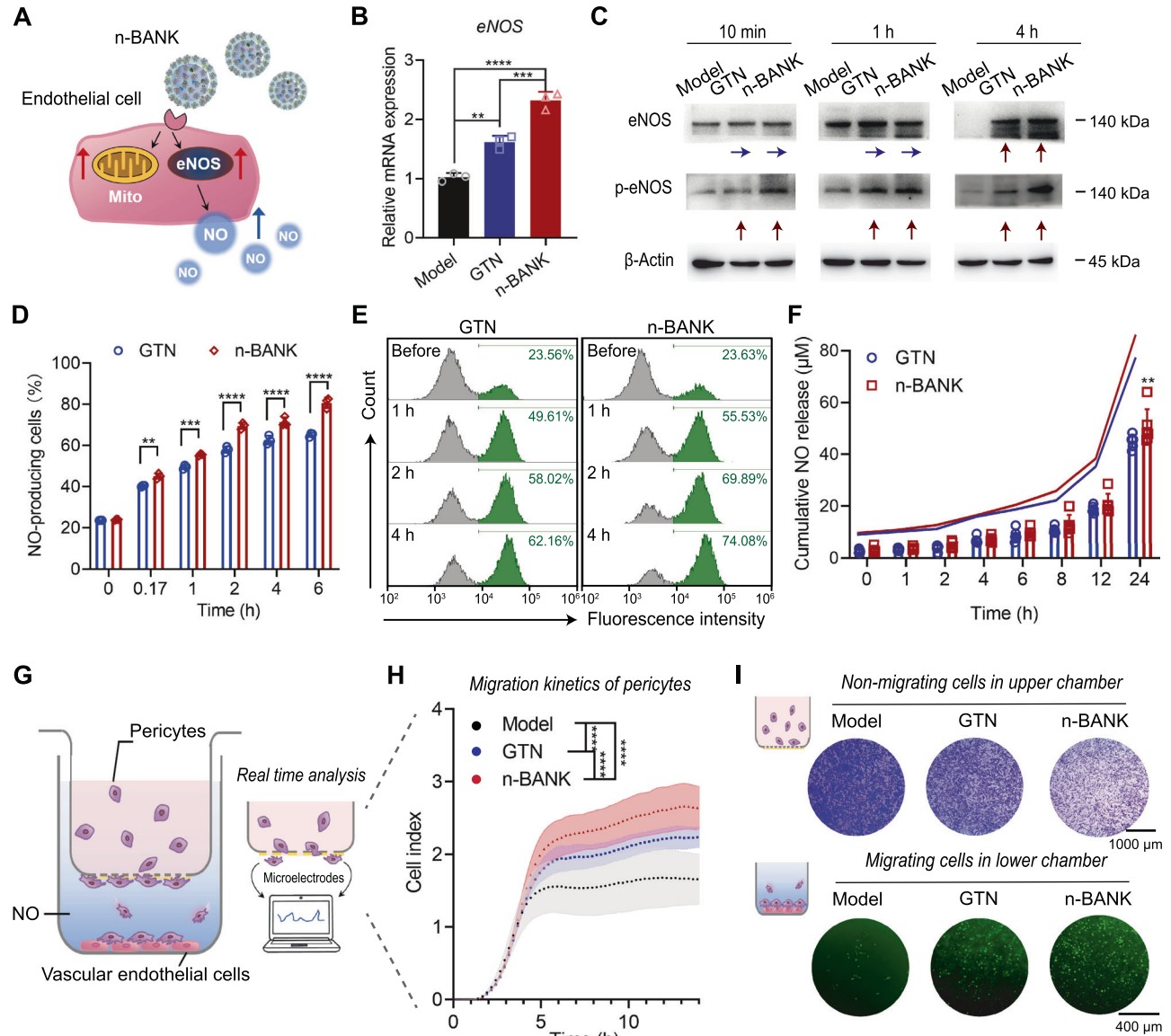

**Fig. 4 | n-BANKs augmented eNOS activity in damaged endothelial cells to facilitate pericyte recruitment. A** Schematic illustration of the promoting effect of n-BANKs on eNOS activation in SVEC4–10 cells. **B** Total RNA isolated from SVEC4–10 cells after exposure to GTN or n-BANKs at 4 h after treatment was examined by real-time qPCR and the mRNA levels of *eNOS* were normalized to the control *GAPDH* mRNA levels ($n = 3$; Model versus GTN, **$P = 0.0015$; Model versus n-BANK, ****$P < 0.0001$; GTN versus n-BANK, ***$P = 0.0006$). SVEC4–10 cells were exposed to 500 μM $CoCl_2$ for 12 h prior to treatment. **C** Western blot of eNOS (phosphorylated and total) in SVEC4–10 cells ($n = 3$). **D, E** Flow cytometry analysis of NO levels in SVEC4–10 cells ($n = 3$; 0.17 h, **$P = 0.0037$; 1 h, ***$P = 0.0003$; 2 h, ****$P < 0.0001$; 4 h, ****$P < 0.0001$; 6 h, ****$P < 0.0001$). SVEC4–10 cells were exposed to 500 μM $CoCl_2$ for 12 h prior to treatment. **F** Release of NO in cell supernatant of SVEC4–10 cells was quantified using Griess reagent ($n = 6$; **$P = 0.0022$). **G** Schematic illustration of real-time cell migration assays. Damaged SVEC4–10 cells were placed in the lower chambers, and the primary pericytes were loaded in the upper chambers. **H** Migration kinetics of pericytes by real-time cell migration assays ($n = 3$; ****$P < 0.0001$). **I** Crystal violet assay and fluorescence imaging revealed pericyte recruitment ability by n-BANKs ($n = 3$). The non-migrating cells in the upper chambers were stained by crystal violet, and the migrating cells in the lower chambers were stained with Calcein-AM. Data are mean ± SD, * is $P < 0.05$, ** is $P < 0.01$, *** is $P < 0.001$, **** is $P < 0.0001$ by one-way or two-way ANOVA test.

n-BANKs (Fig. 4D, E), which displayed a 20.0-fold increase in NO release in 24 h when compared to an untreated control and EVs treated group (Fig. 4F and Supplementary Fig. 9C). Interestingly, the enhancement effect of n-BANKs was even more pronounced than GTN (Fig. 4D–F), likely due to n-BANKs promoting endothelial cell proliferation to further enhance eNOS activity (Supplementary Fig. 9D–E).

To evaluate whether endothelial-derived NO induced by n-BANKs could recruit pericytes efficiently, real-time cell migration assays, as shown in Fig. 4G, were performed. Damaged SVEC4–10 cells untreated or treated with n-BANKs were used to fill the lower chambers. Following upper chamber attachment, primary pericytes from mouse

vessels were loaded in the upper chambers. A porous filter allowing for the migration of pericytes separated the upper and lower chambers. The migration profiles indicated that pericytes displayed more effective migration to the n-BANK treated SVEC4–10 cells compared with the GTN treated or untreated endothelial cells (Fig. 4H). The crystal violet assay and fluorescence imaging revealed that pericyte recruitment to the lower chamber by n-BANK treated SVEC4–10 cells was highly efficient, which showed 2.99-fold and 1.70-fold increases than those of untreated and GTN treated endothelial cells, respectively (Fig. 4I and Supplementary Fig. 10). Conversely, using an eNOS inhibitor $N^G$-nitro-L-arginine methyl ester (L-NAME) to block NO synthesis

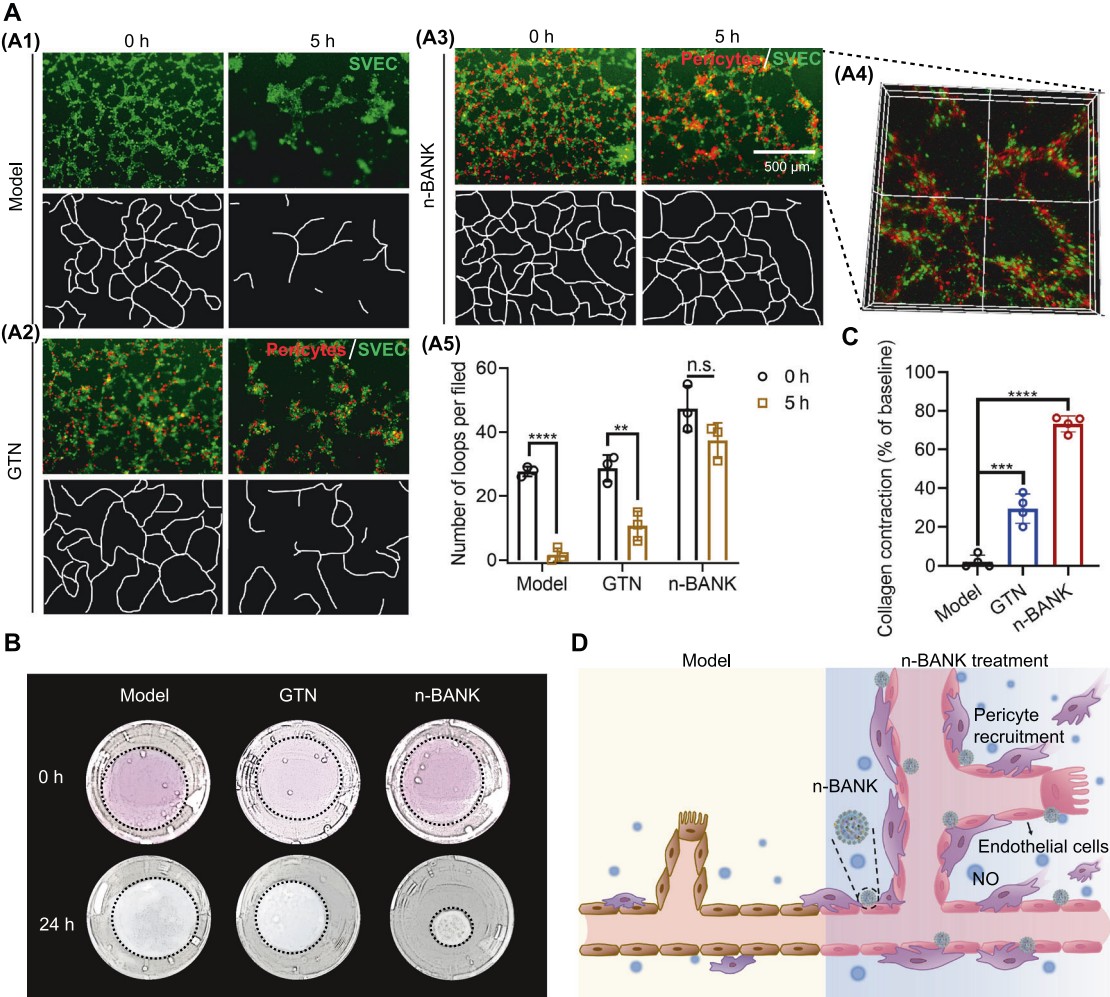

**Fig. 5 | n-BANKs stabilized the vessel-like structures and built functional pericyte-endothelial cell tubes. A** Representative images and quantitative assessment of pericyte recruitment by Matrigel angiogenesis assays at 5 h after endothelial tube formation, which was 9 h after treatment (*n* = 3; Model, ****P < 0.0001; GTN, ***P = 0.0001; n-BANK, n.s. = 0.0586). Pericytes were labeled with DiI (red) and SVEC cells were stained with Calcein-AM (green). Scale bar,

500 μm. **B** Evaluation of functional blood vessel formation by collagen gel contraction assays. **C** Quantification of collagen contraction between pericytes and endothelial cells (*n* = 4; GTN, **P = 0.0012; n-BANK, ****P < 0.0001). **D** Diagram illustrating recruitment of pericytes to vascular endothelium by n-BANKs. Data are mean ± SD, ** is P < 0.01, *** is P < 0.001, **** is P < 0.0001, n.s. is P > 0.05 by one-way or two-way ANOVA test.

in n-BANK treated SVEC4–10 cells markedly reduced recruitment of pericytes to endothelial cells (Supplementary Fig. 11). Together, n-BANKs energized pericyte-endothelial cell interactions by promoting endothelial-derived NO release from endothelial cells to facilitate pericyte recruitment.

## n-BANKs energized pericyte-endothelial cell interactions to build functional vascular networks

Critical for creating a well-functioning vascular network is pericyte recruitment to stabilize immature endothelial cell tubes[18]. Matrigel angiogenesis assay indicated that the number of loops and master junctions of the endothelial tubular network were significantly reduced at 5 h after endothelial tube formation, suggesting regression occurred (Fig. 5A1). However, it is clear that n-BANKs recruited pericytes, stabilized the vessel-like structures and prevented regression (Fig. 5A3). Confocal microscope analysis further revealed that red fluorescent recruited pericytes by n-BANKs integrated with green fluorescent endothelial cells in the pericyte-endothelial cell tubes (Fig. 5A4), which was reminiscent of functional blood vessels in vivo. Additionally, this effect of blocking endothelial cell tube regression was blunted in the GTN treated group, likely due to smaller number of recruited pericytes investing on the endothelial cell tubes (Fig. 5A2,

A5). Next, to determine whether n-BANKs energizing pericyte-endothelial cell interactions were favorable to mature microvascular formation, a collagen gel contraction assay was performed. Endothelial cells and the recruited pericytes were embedded into a 3D collagen matrix and the contractility was monitored. Results showed a 77.30% increase in contractile activity in n-BANK treated group, while GTN treated group induced only 29.49% of gel contraction compared to their original size (Fig. 5B, C). In contrast, endothelial cells without any pericyte investment did not cause any contraction. Collectively, these results raised the possibility that n-BANKs accelerated pericyte recruitment to endothelial cell tubes and subsequently built functional pericyte-endothelial cell tubes (Fig. 5D).

To further investigate whether n-BANKs induced formation of functional pericyte-endothelial cell tubes in vivo, muscle tissues were collected from the severe hindlimb ischemic mice and analyzed for the pericyte recruitment to blood vessels. We used nerve/glial antigen 2 (NG2) proteoglycan as a marker for pericytes in newly formed blood vessels, since it is expressed by microvascular pericytes[39]. Figure 6A showed that there were few NG2-positive pericytes in the ischemic limbs of either untreated or GTN treated mice, indicating the poor pericyte recruitment. However, n-BANK treatment significantly increased the number of NG2-positive pericytes in ischemic limbs.

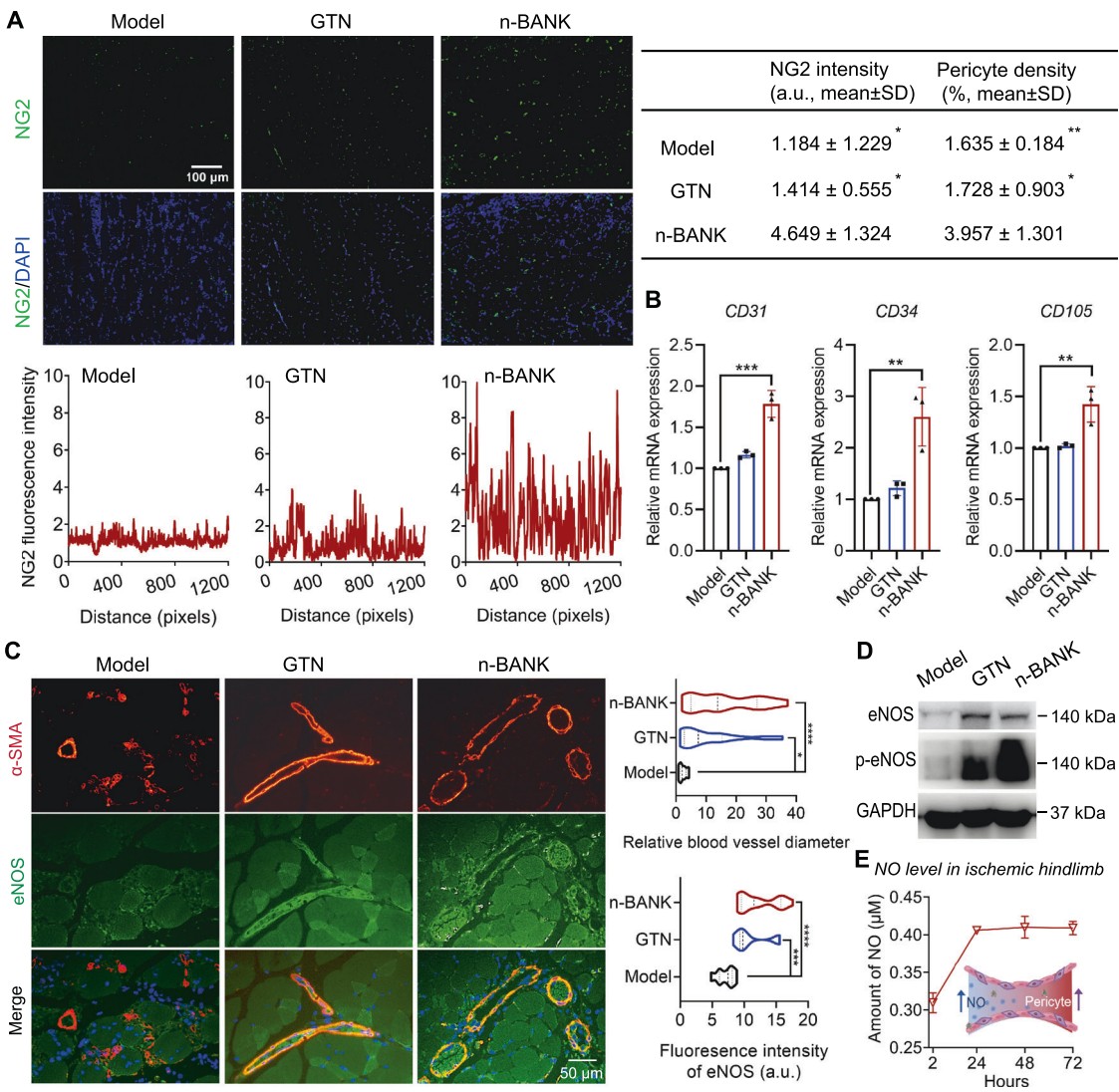

| | NG2 intensity (a.u., mean±SD) | Pericyte density (%, mean±SD) |
|---|---|---|
| Model | 1.184 ± 1.229 * | 1.635 ± 0.184 ** |
| GTN | 1.414 ± 0.555 * | 1.728 ± 0.903 * |
| n-BANK | 4.649 ± 1.324 | 3.957 ± 1.301 |

**Fig. 6 | n-BANKs recruited pericytes for enhancing the maturity of the newly formed blood vessels in CLI mouse model. A** Representative images and quantification showing NG2 expression and pericyte coverage in ischemic muscle sections ($n = 3$; NG2 intensity: Model *$P = 0.0144$, GTN*$P = 0.0194$; Pericyte density: Model **$P = 0.0375$, GTN *$P = 0.0439$). Scale bar, 100 μm. **B** qRT-PCR analysis of endothelial markers (*CD31*, *CD34* and *CD105*) of muscle tissue form ischemic limbs at day 3 after treatment ($n = 3$; *CD31*, ***$P = 0.0001$; *CD34*, **$P = 0.0021$; *CD105*, **$P = 0.0037$). **C** Representative immunofluorescence images and quantification of ischemic gastrocnemius muscle sections stained with eNOS (green) and α-SMA (red) at day 3 after treatment ($n = 17$; Relative blood vessel diameter: Model versus GTN *$P = 0.0212$, Model versus n-BANK ****$P < 0.0001$; Fluorescence intensity of eNOS: Model versus GTN ***$P = 0.0001$, Model versus n-BANK ****$P < 0.0001$). Scale bar, 50 μm. **D** Western blot images showing total and phosphorylated protein levels of eNOS in muscle from ischemic hindlimb ($n = 3$). **E** NO level in ischemic hindlimb treated with n-BANKs ($n = 3$). Data are mean ± SD, * is $P < 0.05$, ** is $P < 0.01$, *** is $P < 0.001$, **** is $P < 0.0001$ by one-way or two-way ANOVA test.

Furthermore, qRT-PCR analysis of ischemic muscle revealed that endothelial markers (*CD31*, *CD34,* and *CD105*) were highly expressed in n-BANK treated ischemic limbs (Fig. 6B). These results suggested that n-BANKs promoted a bicellular vasculogenic population including pericytes and endothelial cells, which is consistent with a typical in vivo microvascular architecture in normal tissues.

Next, we sought to investigate function of blood vessels induced by n-BANKs in a mouse model of severe ischemic hindlimbs. α-smooth muscle actin (α-SMA)-positive cells were used as the index of the mature vasculature. After 3 d of treatment, the animals were sacrificed and the ischemic muscle sections were stained with α-SMA antibody and eNOS antibody, as shown in Fig. 6C. Notably, the mean internal diameter of α-SMA-positive blood vessels in n-BANK treated tissues was significantly larger than that of the untreated control, demonstrating n-BANKs caused marked vasodilation of blood vessels in ischemic tissues. Local vasodilation is thought to be accompanied by augmenting blood flow and delivering nutrients in focal ischemic

tissues. Immunohistochemical analysis further revealed a significant increase in eNOS expression in ischemic muscle tissues of n-BANK treated groups relative to untreated control. These results were consistent with qRT-PCR data in endothelial cell line and western blotting data in ischemic muscle (Fig. 6D and Supplementary Fig. 12). We subsequently compared eNOS-derived NO production in ischemic muscle tissues after n-BANK treatment and control. Interestingly, n-BANKs induced sustainable release of eNOS-derived NO within 7 d after treatment (Fig. 6E). Based on the above data, we herein speculate that n-BANKs induced endothelial cells to produce eNOS-derived NO, and then sustainably pay it out upon pericyte recruitment to create well-functioning mature blood vessels.

### Improvement of ischemic limb salvage by complete revascularization in a severe hindlimb ischemia model

After showing a striking acceleration of functional blood vessel formation induced by n-BANKs, we then evaluated whether they could

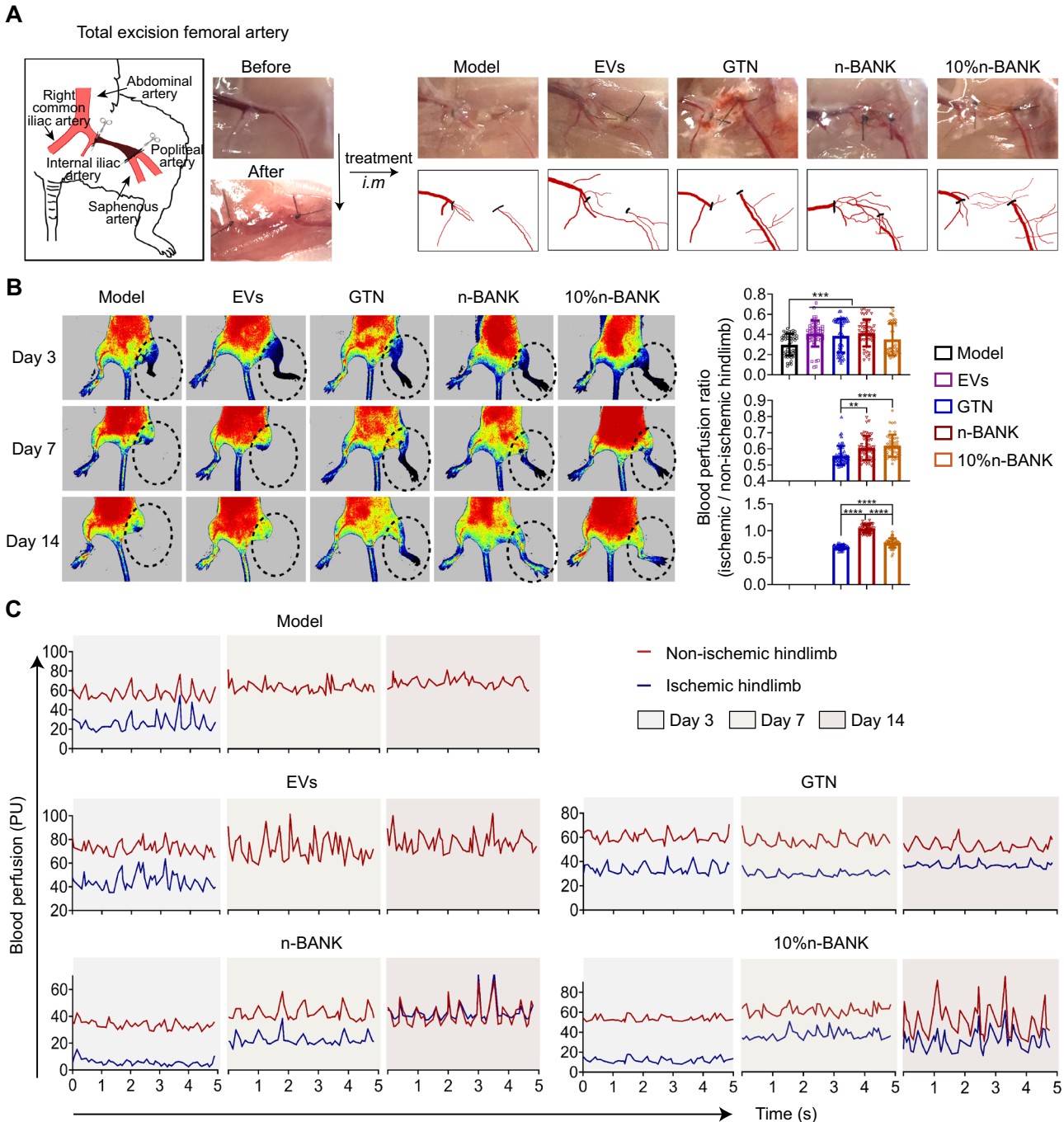

**Fig. 7 | The effect of complete revascularization to normalize blood flow in the ischemic areas. A** Photographs showing new vessel formation across ligation sites of the ischemic hindlimbs at day 3 after treatment. Mice underwent femoral artery ligation to induce hindlimb ischemia. **B** Representative images and quantitative assessment of blood flow after hindlimb ischemia by laser Doppler at indicated day (*n* = 50; Day 3: Model versus EVs ***P = 0.0005, Model versus GTN ***P = 0.0007,

Model versus n-BANK ***P = 0.0002, Model versus 10% n-BANK ***P = 0.0003; Day 7: GTN versus n-BANK **P = 0.0017, GTN versus 10% n-BANK ****P < 0.0001; Day 14: ****P < 0.0001). **C** Time course of blood flow recovery by laser Doppler imaging in hindlimb of mice treated with different formulations. Data are mean ± SD, ** is *P* < 0.01, *** is *P* < 0.001, **** is *P* < 0.0001 by one-way ANOVA test.

confer revascularization to decrease the amputation rate following severe ischemic injury. A mouse model for severe hindlimb ischemia was established by ligating the proximal and distal femoral artery in the left limb, and then excising the arterial segment between the ligatures. The right limb served as the contralateral non-occluded control. Notably, n-BANK treatment induced more extensive microvessels around the ligation sites of femoral artery and saphenous artery that bridged the gap between the ligation sites after 3 d following severe ischemic injury (Fig. 7A). Laser Doppler imaging was

employed to assess the blood perfusion in the severe ischemic hindlimbs. As shown in Fig. 7B, C and Supplementary Movies 1–5, no obvious blood-flow signal was monitored in ischemic hindlimbs of mice in all groups at 3 d after severe ischemic injury. Strikingly, both n-BANK treated and low-dose n-BANK treated mice showed marked improvement in blood perfusion at 7 d and achieved 103.8 ± 7.6% and 77.3 ± 8.4% restoration of blood flow to the non-occluded limbs at 14 d after CLI, respectively, which was significantly higher than other control groups.

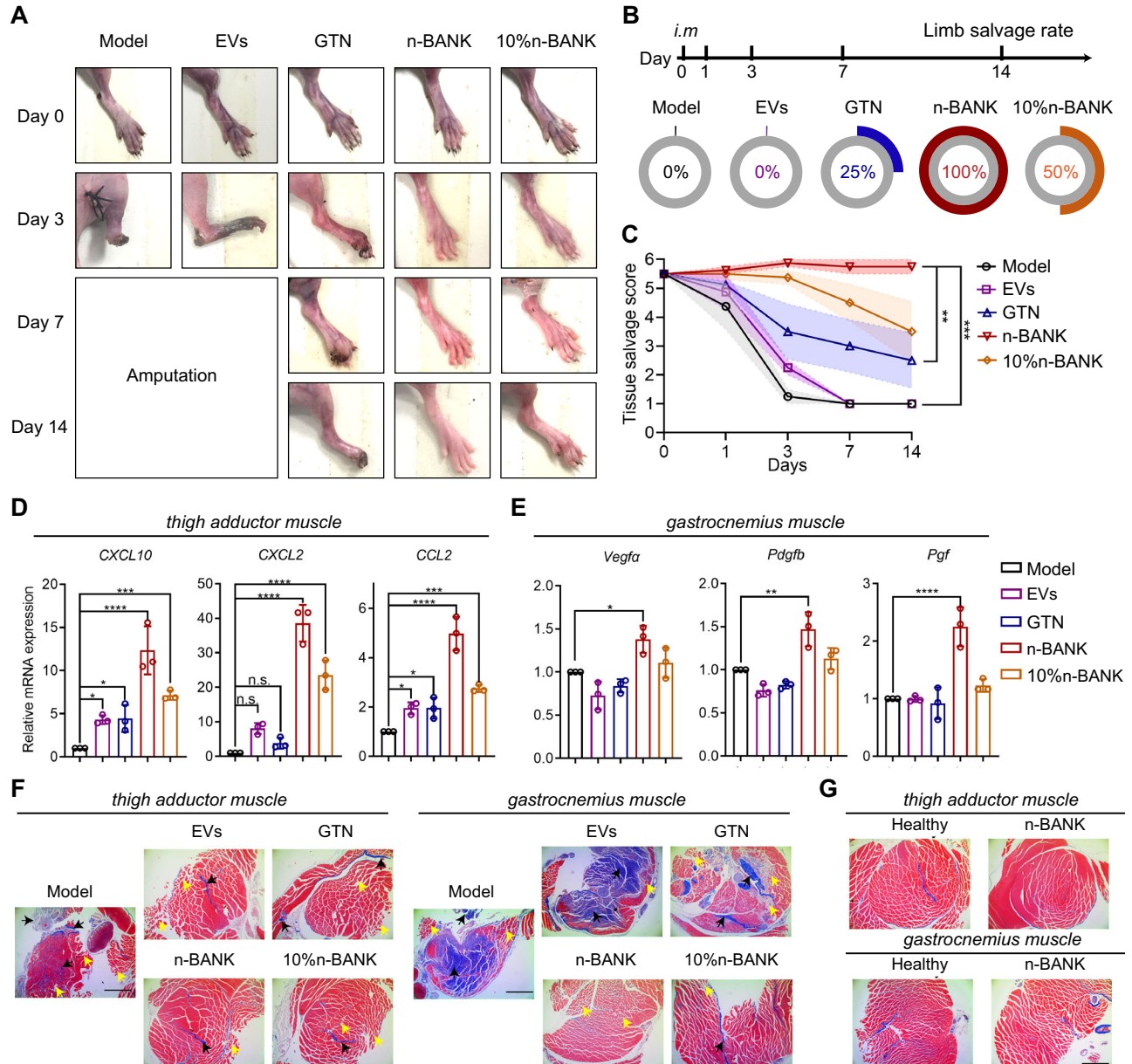

**Fig. 8 | n-BANKs improved limb salvage following severe ischemic injury.**
**A** Representative serial photographs of ischemic hindlimbs at days 0, 3, 7, and 14 after treatment. **B** Limb salvage rate and (**C**) tissue salvage sore of ischemic hindlimbs in different treatment groups (*n* = 4; n-BANK versus Model ***P* = 0.0004, n-BANK versus EVs ***P* = 0.0004, n-BANK versus GTN **P* = 0.0094). **D** Different genes related to collateral artery remodeling on thigh adductor muscle and (**E**) different blood flow recovery related genes on gastrocnemius muscle harvested from ischemic hindlimbs at day 3 (*n* = 3; *CXCL10*: Model versus EVs *P* = 0.0419, Model versus GTN *P* = 0.0410, Model versus n-BANK ****P* < 0.0001, Model versus

10% n-BANK ****P* = 0.0006; *CXCL2*: ****P* < 0.0001; *CCL2*: Model versus EVs *P* = 0.0400, Model versus GTN *P* = 0.0384, Model versus n-BANK ****P* < 0.0001, Model versus 10% n-BANK ***P* = 0.0008; *Vegfα*: *P* = 0.0181; *Pdgfb*: **P* = 0.0014; *Pgf*: ****P* < 0.0001). **F** Masson's trichrome staining of histological ischemic hindlimb sections at day 3 after surgery (*n* = 3). **G** Masson's trichrome staining of muscle sections from healthy mice and n-BANK treated mice at day 14 after surgery. Light blue staining for collagen indicates fibrosis, and muscle fibers are stained red (*n* = 3). Scale bar, 1000 μm. Data are mean ± SD, * is *P* < 0.05, ** is *P* < 0.01, *** is *P* < 0.001, **** is *P* < 0.0001 by one-way ANOVA test.

Next, we sought to investigate whether severe ischemic limbs would benefit from timely restoration of blood flow to the ischemic area by n-BANK treatment. As shown in the representative photographs and the corresponding limb salvage scoring (Fig. 8A–C), all CLI mice in untreated model group were observed complete limb loss at 3 d after surgery, likely due to the restricted blood flow resulting in a lack of oxygen to the skeletal muscles. The mice treated with activated EVs showed grossly extensive forefoot necrosis and total toe amputation at 3 d, and then complete limb loss within 7 d. GTN treated group also displayed a high frequency of necrotic toes at 3 d and a rate of spontaneous limb loss of 75% after 14 d. EVs were not effective against

ischemic injury in vivo, most likely due to that EVs without NO-nanocages could not recruit pericytes effectively to create well-functioning mature blood vessels. Notably, n-BANK treatment achieved limb salvage in all experimental mice within 14 d following severe ischemic injury. Five of six mice with complete revascularization fully recovered their legs and the other one showed minor necrosis. Additionally, 1/10th dose of n-BANK treated mice had incomplete revascularization and showed 50% limb salvage after 14 d, which was also superior to either activated EVs or GTN.

We also sought to investigate whether n-BANKs achieving complete revascularization could control muscle recovery from ischemia

in vivo. Quantitative analysis of cytokines and chemokines, which are associated with tissue regeneration, from gastrocnemius and thigh adductor muscle extracts were performed at 3 d after treatment. As Fig. 8D demonstrated, the three chemokines, known as the markers for collateral remodeling[40], were upregulated by 12.3-fold (*CXCL10*), 38.6-fold (*CXCL2*), and 5.0-fold (*CCL2*), respectively, in the thigh adductor muscle treated with n-BANKs than those of the ischemic control. Moreover, n-BANK treatment was observed a pronounced up-regulation of several cytokines (*Vegfα, Pdgfb, Pgf, Hbegf, Tgfb3*), which exerted the important role in regeneration in gastrocnemius muscle, compared with the ischemic control (Fig. 8E and Supplementary Fig. 13A). Additionally, this treatment also caused upregulation of mitochondrial biogenesis-related genes and downregulation of inflammatory response-related genes in injured skeletal muscle (Supplementary Figs. 13B, 14, and 15), implicating recovery from ischemia-induced muscle damage.

To ascertain the reduction extent of ischemia-induced muscle injury in the experimental mice receiving n-BANKs and other treatment, Masson's trichrome staining histological analyses were performed at 3 d and 14 d after treatment. The untreated ischemic mice failed to maintain normal muscles and had irregular deposition of collagen fibers (blue staining, black arrows) with loose basket-weave morphology (red staining, yellow arrows) at 3 d following severe ischemic injury (Fig. 8F). When compared to untreated ischemic mice, all the treated groups markedly reduced muscle degeneration, while only the n-BANK and low-dose n-BANK treated groups showed substantial reduction of irregular deposition of collagen fibers. Furthermore, n-BANK treated mice displayed dramatic improvement associated with muscle degeneration and fibrosis and showed the similar muscle morphology as normal muscles at 14 d (Fig. 8G and Supplementary Figs. 13C and 16). Taken together, these results demonstrated that n-BANKs sufficed to induce therapeutic vessel growth and achieve complete revascularization, resulting in marked improvement in ischemic limb salvage and muscle regeneration.

### n-BANKs promoted motor function recovery following severe ischemic injury

Complete revascularization and muscle regeneration are associated with improved clinical prognosis following ischemic injury[41,42]. Thus, the recovery of motor function after severe hindlimb ischemic injury was investigated. Motor function of the healthy group and limb-salvage groups (n-BANK, 10% n-BANK, and GTN) was evaluated including grasping reflex, hindlimb grip strength and hindlimb suspension. We utilized movies to record the grasping ring test of experimental mice at 14 d after treatment. The n-BANK treated CLI mice displayed a strong grasp reflex to grab a blunt ring when the ring was placed against the palm of injured hindlimb. These mice were also able to maintain their grip and hold their body weight for a few seconds, which was comparable to healthy controls (Fig. 9A and Supplementary Movies 6, 7). Additionally, 10% n-BANK treated CLI mice also recovered the grasping function of ischemic hindlimb to some extent, but their hindlimb grip strength could not hold their body weight (Supplementary Movie 8). By contrast, CLI mice with only GTN treatment showed hindlimb weakness and a loss of fine motor skills, such as grasping, as demonstrated by the grasping reflex disappeared and a significant deficit in grip strength (Supplementary Movie 9).

Mice tend to walk with balanced symmetric gaits, and deviations from this spatiotemporal pattern can help identify unilateral limb injuries[43]. Thus, the gait analysis was performed in limb-salvage groups at 14 d after treatment (Fig. 9B). Interestingly, in agreement with perfusion data, the test gait parameters including stride distance, stance length, intra-step distance and overlap distance were not significantly different between n-BANK treated CLI mice and healthy control, suggesting that n-BANKs were able to restore full motor function of mice following severe ischemia injury (Fig. 9C, D). By contrast, GTN treated

group showed motor function deficits in ischemic hindlimb that the stride distance, stance length and intra-step distance were markedly reduced, while overlap distance was increased, due to that those uncured ischemic mice shifted body weight to the intact limbs to partially compensate by postural adjustments.

Additionally, there has been a growing realization that severe limb ischemia and limb ischemia-reperfusion initiate the inflammatory cytokine cascade, leading to remote organ failure and adverse outcomes[44]. We, therefore, assessed the liver/kidney injury at 14 d after treatment. Indeed, severe limb ischemia caused marked elevation of alanine aminotransferase (ALT), a clinical tool for detecting liver injury, and lactate dehydrogenase (LDH), a marker of ischemic damage (Fig. 9E). Concomitantly, creatinine (Cre) and blood urea nitrogen (BUN) levels rose significantly in mice with ischemia injury, suggesting decline in kidney function (Fig. 9F). As expected, the levels of ALT, LDH, Cre, and BUN returned to baseline levels of healthy mice after n-BANK treatment, demonstrating recovery of liver and kidney function in ischemia-injured mice.

Collectively, these data demonstrated that n-BANK treatment achieving complete revascularization was able to restore full motor function of CLI in a dose-dependent manner and prevent remote organ injury.

## Discussion

CLI is a major public health concern worldwide with increased numbers of cases and a high risk for major amputation[9]. Generation of a well-functioning vascular network for revascularization is critical to the clinical success of CLI treatment and has not yet been completely demonstrated. Functional microvascular regeneration depends on endothelial cell morphogenesis into a tubular network, following by stabilization of the assembling structures by recruited pericytes. Most studies of therapeutic angiogenesis have concentrated mainly on the activation of endothelial cells by angiogenic molecules, such as members of the VEGF or FGF family. However, endothelial cells potently activated by these angiogenic molecules tend to form immature nascent vessels, possibly resulting in widespread micro-vascular leakage with subsequent edema formation[18,45]. Pericytes with contractile and relaxant properties are functionally significant, but receive considerably less attention than endothelial cells[16,17]. In this study, n-BANKs, constructed by activated EVs decorated with NO-nanocages, could store NO in endothelial cells and pay it out upon pericyte recruitment to create well-functioning blood vessels for complete revascularization.

Evidence shows that eNOS is important for pericyte recruitment[20]. We noted here that the ischemic microenvironment of CLI had strongly deleterious effects on eNOS activity and investment of pericytes in the microvessels. n-BANKs shuttling NO-nanocages could stimulate NO synthesis by increasing the phosphorylation of eNOS, and store eNOS-derived NO in endothelial cells. Moreover, the activated EVs in n-BANKs containing multiple angiogenic factors rescued mitochondrial dysfunction and promoted proliferation of endothelial cells, which could achieve sustainable NO production. One of the most interesting findings of this study was that n-BANK induced NO was released from endothelial cells to facilitate pericyte recruitment. The recruited pericytes prevented regression of nascent endothelial tubes, resulting in stabilization of the vascular structures. By contrast, the nascent endothelial tubes without pericyte investment were unstable and prone to regression over time.

Of note, the density of pericytes in newly formed vessels was 2.42 and 2.29-fold higher in n-BANK treated CLI mice, when compared to untreated and GTN treated groups, respectively. Subsequent investment by a significantly great number of NG2-positive pericytes to the nascent endothelial tubes led to build functional pericyte-endothelial cell tubes, which conferred vessel contractility and relaxation. In contrast, endothelial cell tubes without pericyte investment did not

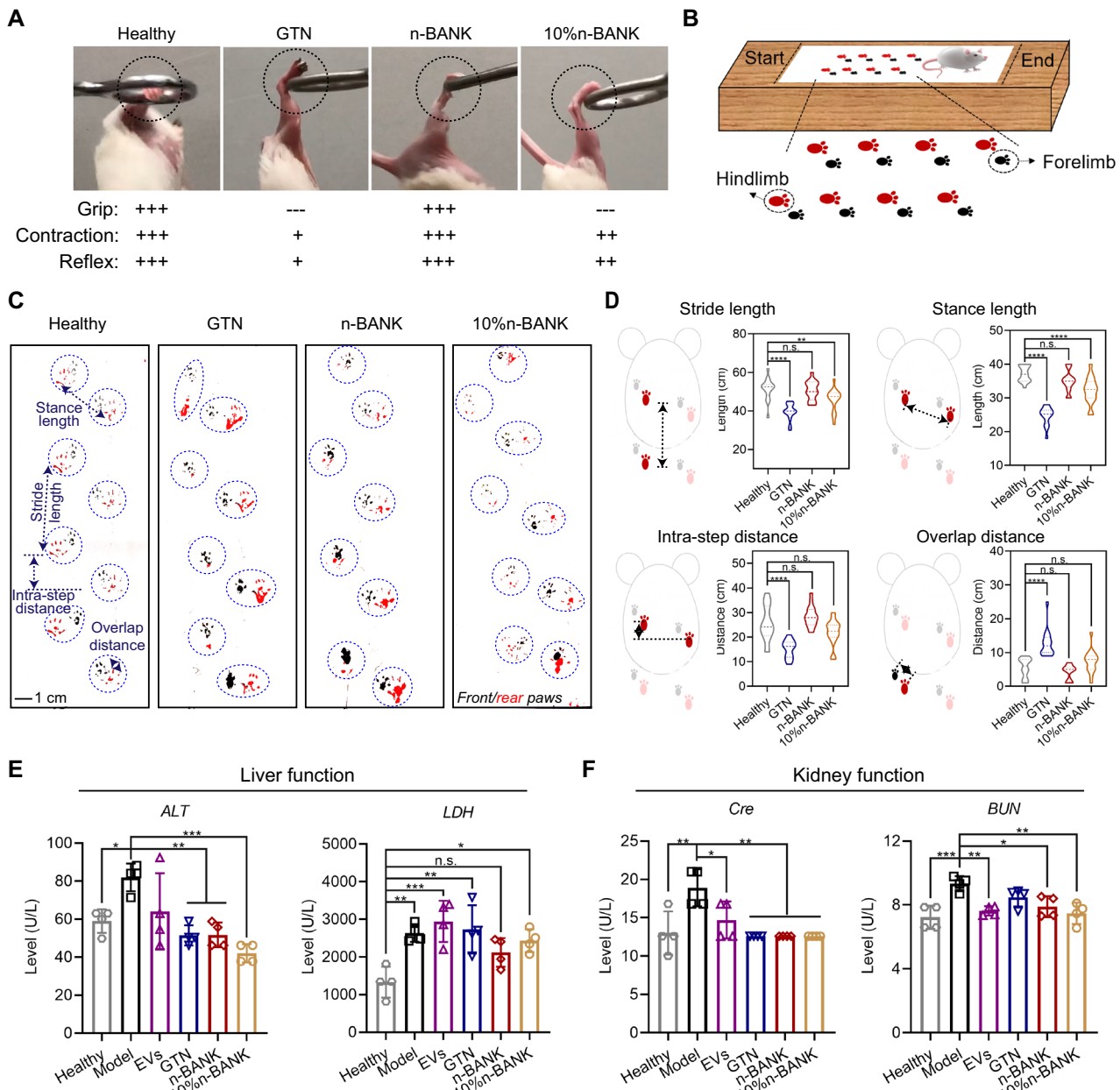

**Fig. 9 | Restoration of full motor function of severe ischemic hindlimbs after n-BANK treatment in a dose-dependent manner. A** Evaluation of hindlimb motor function using the grasping ring test at day 14 after treatment. **B** Schematic of painted footprint experiment used to study gait: forelimbs were painted black and hindlimbs were painted red. **C** Representative footprints from mice in each group at day 14 after treatment. **D** Bar graphs showing differences in stance width and stride parameters among these groups (*n* = 16; Stride length: Healthy versus GTN ****P < 0.0001, Healthy versus 10% n-BANK **P = 0.0098; Stance length: ****P < 0.0001; Intra-step distance: ****P < 0.0001; Overlap distance: ****P < 0.0001). **E**, **F** Serum biochemical parameters of liver function and kidney function were

analyzed (*n* = 4; ALT: Model versus Healthy *P = 0.0414, Model versus GTN **P = 0.0045, Model versus n-BANK **P = 0.0048, Model versus 10% n-BANK **P = 0.0003; LDH: Healthy versus Model **P = 0.0064, Healthy versus EVs ***P = 0.0008, Healthy versus GTN **P = 0.0033, Healthy versus 10% n-BANK *P = 0.0248; Cre: Model versus Healthy **P = 0.0026, Model versus EVs *P = 0.0408, Model versus GTN **P = 0.0013, Model versus n-BANK **P = 0.0013, Model versus 10% n-BANK **P = 0.0013; BUN: Model versus Healthy ***P = 0.0009, Model versus EVs **P = 0.0070, Model versus n-BANK *P = 0.0277, Model versus 10% n-BANK **P = 0.0033). Data are mean ± SD, * is P < 0.05, ** is P < 0.01, *** is P < 0.001, **** is P < 0.0001, n.s. is P > 0.05 by one-way ANOVA test.

show any contractile activity in vitro. Moreover, in vivo experiments also demonstrated that n-BANKs energized pericyte-endothelial cell interactions via NO for successful development of a functional vascular network and further improvement of NO-dependent vasodilation. The functional blood vessel networks were essential to provide oxygen and nutrients to the ischemic tissues.

Strikingly, the nascent and mature vessels induced by n-BANKs provided adequate blood flow to the ischemic tissues, resulting in complete revascularization that is thought to be associated with a

better prognosis for ischemic injury. They also demonstrated the potential to greatly ease the inflammatory storm triggered by severe ischemic injury. Consequently, n-BANK treatment markedly improved ischemic limb salvage and even restored the motor function within 14 d following severe ischemic injury. Conversely, EVs that delivered angiogenic factors but not capable of boosting eNOS-derived NO production, failed to show meaningful therapeutic benefit and amputation rates remained high. Thus, given diabetes, hypertension and dyslipidemia raise the risk of developing

CLI, which is likely to be an increasingly important issue, the n-BANKs may provide a promising strategy for CLI treatment and ultimately addressing the problem of high amputation rates of CLI. Although the n-BANK therapy holds promise for CLI treatment, future challenges still exist in translating it to clinical practice. These mainly include the use of EVs from human stem cells to construct n-BANKs as therapeutic resources and further exploration of the n-BANK stimulated eNOS-NO signaling pathways in pericyte-endothelial interactions. Overcoming these challenges would make n-BANKs an effective platform for clinical applications.

## Methods

### Experimental animals

All the animal experiments conducted strictly observed the Guiding Principles for the Use of Laboratory Animals and were approved by the Institutional Animal Care and Use Committee of the Sun Yat-sen University, ethical approval number No.44008500026874. Female BALB/c mice at 8–10 weeks of age were purchased from the Laboratory Animal Center, Sun Yat-sen University. Efforts were made to minimize animal suffering, and the necessary sample size was calculated before the experiments to reduce the number of animals used per condition. All the animals were housed under specific pathogen free (SPF) barrier environment (20–26 °C and at 40–70% humidity), under the 12 h/12 h dark/light cycle. All mice had free access to standardized food and water. Isoflurane (2%, 0.5 L/min) were used in animal euthanasia practice.

### Reagents

Resveratrol, 4′,6-Diamidino-2-phenylindole dihydrochloride (DAPI), 1,10-phenanthroline, dimethyl sulfoxide (DMSO), matrigel and 3-(4,5-dimethylthiazol-2-yl)-2,5-diphenyl tetrazolium bromide (MTT) were purchased from Sigma-Aldrich (USA). Cobalt(II) chloride hexahydrate ($CoCl_2 \cdot 6H_2O$) was purchased from Guangzhou chemical reagent factory (Guangzhou, China). Glyceryl trinitrate (GTN) injection (5 mg/mL) was obtained from Beijing Yimin Pharmaceutical Co., Ltd (Beijing, China). Pierce bicinchoninic acid (BCA) protein assay kit was supplied by Thermo Scientific (USA). Roswell Park Memorial Institute (RPMI)-1640 medium, Dulbecco's modified Eagle medium (DMEM), Dulbecco's modified Eagle medium/nutrient mixture F-12 (DMEM/F12), fetal bovine serum (FBS), trypsin and penicillin/streptomycin were purchased from Gibco (Canada). Recombinant mouse macrophage colony stimulating factor (M-CSF) was purchased from MedChem Express (USA). TRIzol reagent was the products of Invitrogen (Carlsbad, USA). PrimeScript™ RT reagent kit with gDNA eraser (Perfect Real Time) and TB Green™ Premix Ex Taq™ II (TliRNaseH Plus) were purchased from Takara Bio (Shiga, Japan). 4% paraformaldehyde solution were supplied by Leagene Biotechnology (Beijing, China). 1,1′-dioctadecyl-3,3,3′,3′-tetramethylindocarbocyanine perchlorate (DiI), 1,1′-dioctadecyl-3,3,3′,3′-tetramethylindodicarbocyanine,4-chlorobenzenesulfonate salt (DiD), mitochondrial membrane potential assay kit, calcein acetoxymethyl ester (Calcein-AM), crystal violet staining solution, radio immunoprecipitation assay (RIPA) lysis buffer, phenylmethanesulfonyl fluoride (PMSF), nitric oxide assay kit, NO-specific, cell and tissue lysis buffer for nitric oxide assay, fluorescent probe 4-amino-5-methylamino-2′,7′-difluorofluorescein diacetate (DAF-FM DA), primary and secondary antibody dilution buffer and loading buffer were obtained from Beyotime Biotechnology (Shanghai, China). Immobilon western chemiluminescent horseradish peroxidase (HRP) substrate and polyvinylidene fluoride (PVDF) membranes were purchased from Millipore (USA). Rat tail tendon collagen type I was purchased from Solarbio (Beijing, China). Anti-vascular endothelial growth factor α (VEGFα) antibody (rabbit, Cat# ab214424, dilution 1:1000), anti-von Willebrand factor (vWF) antibody (rabbit, Cat# ab174290, dilution 1:1000), anti-endothelial nitric oxide synthase (eNOS) antibody (rabbit, Cat# ab199956, dilution 1:1000),

anti-phospho-eNOS antibody (rabbit, Cat# ab215717, dilution 1:1000), anti-beta actin antibody (rabbit, Cat# ab8227, dilution 1:1000), anti-beta tubulin antibody (rabbit, Cat# ab6046, dilution 1:500), anti-glyceraldehyde-3-phosphate dehydrogenase (GAPDH) antibody (rabbit, Cat# ab9485, dilution 1:2500), anti-rabbit IgG H&L (HRP) (goat, Cat# ab6721, dilution 1:2000) for western blot assays were supplied by Abcam (Britain). Cell staining buffer, APC anti-mouse/human CD11b antibody (Cat# 101211, dilution 1:80), FITC anti-mouse/rat CD29 antibody (Cat# 102205, dilution 1:50) and FITC anti-mouse Ly-6A/E (Sca-1) antibody (Cat# 122505, dilution 1:50) for flow cytometry assays were supplied by BioLegend (San Diego, USA). APC-R700 rat anti-mouse CD45 antibody (Cat# 565478, dilution 1:80) for flow cytometry assays was obtained from BD Biosciences (USA). Anti CD31 rabbit antibody (Cat# GB11063-2-100, dilution 1:200), anti-nerve/glial antigen (NG2) rabbit antibody (Cat# GB115534-100, dilution 1:200), anti-alpha smooth muscle actin (α-SMA) rabbit antibody (Cat# GB111364-100, dilution 1:300), anti-eNOS mouse antibody (Cat# GB12086-100, dilution 1:500), FITC conjugated goat anti-rabbit IgG (H + L) (Cat# GB22303, dilution 1:100), Cy3 conjugated goat anti-rabbit IgG (H + L) (Cat# GB21303, dilution 1:200) and FITC conjugated goat anti-mouse IgG (H + L) (Cat# GB22301, dilution 1:100) for immunofluorescence assays were purchased from Servicebio (Wuhan, China).

### Cell lines

SVEC4–10 cells (CRL-2181) were purchased from the American Tissue Culture Collection. SVEC4–10 cells were cultured in DMEM medium supplemented with 10% fetal bovine serum and 1% penicillin–streptomycin, and kept in a 37 °C humidified incubator (Thermo, USA) with 5% $CO_2$.

### MSC isolation and culture

MSCs were isolated from the bone marrow of BALB/c mice at 8–10 weeks of age. Firstly, following the sacrifice of mice, the tibias and femurs were removed and cleaned from connective tissue. Marrow was flushed out of the epiphysis cut bones, followed by marrow disintegration to cell suspension by gradual passage through a series of needles (19 G, 21 G, 23 G and 25 G). Finally, cells were suspended in DMEM/F12 medium supplemented with 10% fetal bovine serum and 1% penicillin–streptomycin. Suspended marrow cells were plated in a 100-mm dish and cultured at 37 °C in a humidified atmosphere containing 95% air and 5% $CO_2$. The nonadherent cells were removed after 72 h of culture and cells were passaged when they were 90% confluent by treating them with 0.25% trypsin containing 0.02% EDTA. MSCs were assessed by cytofluorimetric analysis for the expression of the typical markers Sca-1 and CD29, and the absence of the hematopoietic markers CD45 and CD11b. MSCs were used in the experiments only after 2 to 4 expansion passages, to ensure depletion of monocytes/macrophages.

### Isolation of EVs

FBS for cell culture was prepared by overnight ultracentrifugation at 120,000 g, 4 °C as described in reference[46]. $2 \times 10^8$ cells plated in a 150 mm dish were allowed to reach 90% confluence. The conditioned medium was collected and the viability of the cells (>95%) was confirmed with trypan blue exclusion test. Nanovesicles were harvested by sequential centrifugation as previously described[34] with some modifications. Firstly, conditioned medium was centrifuged at 500 g (Avanti J-26S XPI with JA-25.50 rotor, Beckman) for 60 min at 4 °C to remove cells and cell debris. The supernatant was further centrifuged at 2500 g (Avanti J-26S XPI with JA-25.50 rotor, Beckman) for 60 min at 4 °C to remove apoptotic bodies. Subsequently, the supernatant was transferred to a new tube for another centrifugation step for 60 min at 20,000 g (Avanti J-26S XPI with JA-25.50 rotor, Beckman) to collect EVs. Activated MSCs were obtained by incubating MSCs with resveratrol (0.25 µM) for 24 h.

## Preparation and characterization of n-BANKs

GTN solution (5 mg/mL) and albumin solution (1 mg/mL) were added into 100 μL of PBS at room temperature overnight to obtain NO-nanocages. Besides, the EVs were stirred with phosphatidylcholine at the mass ratio of 5:1 for 30 min at 37 °C. Then, NO-nanocages were mixed with the EVs for 60 min at 37 °C. Finally, the mixture was centrifuged at 20,000 g for 60 min at 4 °C to collect n-BANKs, which then were resuspended in sterile PBS (100 μL).

For the loading capacity of GTN in n-BANKs, the supernatant in the final step was collected to measure the unloaded GTN. The supernatant then added into acetone at a final concentration of 70% (vol/vol) at 4 °C for 2 h. The mixture was centrifuged at 12,000 g for 30 min at 4 °C to remove the precipitation of proteins and PC. The GTN concentration was determined by absorption at 211 nm using UV spectrophotometer (UV-2600, Shimadzu, Japan). A calibration curve was established using different concentrations of free GTN in 70% acetone (vol/vol). GTN encapsulation efficiency (EE%) = [1 − mass of free GTN (mg)/mass of total GTN (mg)] × 100%.

For transmission electron microscopy (TEM) imaging, the isolated EVs and n-BANKs were mixed with paraformaldehyde at a final concentration of 4% wt/vol. After incubation at room temperature for 20 min, a drop of sample (100 μL) was added on a sheet of parafilm. A 300-mesh copper grid support film (Electron Microscopy Sciences, PA, USA) was placed on the sample liquid drop (membrane side down) to allow the membranes adsorption for 20 min. Next, the grid was washed three-times in a 100 μL drop of distilled water for 2 min, and then was transferred to a 100 μL drop of uranyl acetate solution for negative staining for 10 min. After washing away with unstained uranyl acetate, the grid was left to air-dry at room temperature. And, the sample was observed under JEM-1400 electron microscope (JEOL, Tokyo, Japan). Particle sizes of EVs and n-BANKs were measured by Nanosight NS300 system (Malvern, UK), and zeta potentials were detected by Zetasizer Nano-2S (Malvern, UK).

## Molecular docking

Molecular docking analysis was performed to study the interaction between GTN and albumin. The 2D structure of GTN was generated by ChemDraw, then converted to 3D structure using Maestro LigPrep in OPLS3 force field, version 2.1.207. The X-ray structure of bovine serum albumin (PDB code: 4JK4) was obtained from the Protein Data Bank (http://www.rcsb.org/pdb). The Grid files were obtained following the standard procedure recommended by Schrödinger. The 3D structure of ligand was docked flexibly using Glide in XP mode, and other docking parameters were set to default values. Ten predicted poses were obtained during the docking process, and the docking score was calculated for each pose. The image files were generated by PyMOL (Version 1.7).

## Surface plasmon resonance analysis

Affinity of PC and albumin was evaluated using Biacore 8 K (GE healthcare, Sweden). The CM5 sensor chip was used as the stationary phase. The albumin solution was added containing 10 mM NaAc (pH 4.5) for 10 min at 10 μL/min flow rate, followed by 1 M ethanolamine hydrochloride to block the chip. For the binding studies, PBS (pH 7.4) containing 5% DMSO was used as the running buffer. A series of PC at different concentrations (31.25, 62.5, 125, 250, and 500 μM) were injected at a flow rate of 30 μL/min at 25 °C. The SPR signal changes were calculated to evaluate the binding profiles.

## Fluorescence spectroscopy

Fluorescence spectra of albumin, NO-nanocage and n-BANKs were recorded on a fluorescence spectrometer (Fluoromax-4, HORIBA, USA) equipped with a plotter unit and a quartz cell (1 × 1 cm). The excitation and emission slit widths were fixed at 5 nm. The excitation wavelength was set at 285 nm, and the emission spectra were read at 300–500 nm.

## Fluorescence imaging of n-BANKs

To confirm the structure of n-BANKs, EVs were prelabeled by DiI and NO-nanocages were prepared with fluorescein 5-isothiocyanate (FITC)-conjugated bovine serum albumin. The fluorescent 3D imaging was performed on a laser scanning confocal microscope (AXR, Nikon, Japan) with a CFI Plan Apochromat Lambda 100×H (NA1.45) objective using excitation at 488 nm for FITC-conjugated NO-nanocages and 549 nm for DiI-labeled EVs.

## Succinate dehydrogenase (SDH) activity

Cells were incubated for 24 h at 5000 cells/well using 96-well plates (Costar, USA). When reaching to 80% confluence, the cells were incubated in the present of 500 μM cobalt chloride (CoCl$_2$) for 12 h, to mimic hypoxic/ischemic conditions. Then, the cells were exposed to varying concentrations of drugs for another 6 h. To determine the SDH activity of cells, 20 μL of MTT solution (5 mg/mL, in phosphate buffered saline) was added to 100 μL of medium in each well of the 96-well plate. The plate was placed in a cell culture incubator until purple precipitates were clearly visible (approximately 4 h). Then, 150 μL of DMSO was added. The absorbance in each well was measured at 570 nm with a microplate spectrophotometer (Epoch, Bio-Tek, USA).

## Mitochondrial membrane potential analysis

SVEC4–10 cells were seeded at 150,000 cells per well in a 12-well plate for 24 h. When reaching to 80% confluence, cells were pretreated with 500 μM of CoCl$_2$ for 12 h prior to nanovesicles (10 μg/mL), n-BANKs (10 μg/mL activated nanovesicles) or PBS for 24 h. Mitochondria were stained with 5,5′,6,6′-tetrachloro-1,1′,3,3′-tetraethylbenzimidazolocarbocyanine iodide (JC-1). JC-1 accumulation in the mitochondria depends on the mitochondrial membrane potential (ΔΨm). JC-1 color alters reversibly from green (J-monomer, at low JC-1 concentration) to red (J-aggregates, at high JC-1 concentration) with increasing ΔΨm. Cells were exposed for 30 min to 2 μM JC-1 in cell culture medium at 37 °C, washed twice and incubated for additional 30 min. Then the cells were harvested and resuspended in 500 μL cell culture medium for FACS measurements (CytoFLEX-S, Beckman, USA). The data are the mean fluorescent signals from 10,000 cells analyzed.

## Quantitative real-time polymerase chain reaction (qRT-PCR)

Total RNA was isolated from frozen tissue or cell line samples using TRIzol Reagent according to the manufacturer's protocol. The RNA concentration and quality were measured using an optical NanoDrop 2000 spectrophotometer (Thermo Fisher Scientific). Total RNA (1000 ng) was transcribed with a High-Capacity cDNA Reverse Transcription Kit (TaKaRa). Quantitative real-time PCR was performed on a Bio-Rad MyiQ Real-Time PCR Detection System (Bio-Rad Laboratories). IQ SYBR Green Supermix (Bio-Rad Laboratories) was used in a 10 μL reaction volume. The expression of individual genes was normalized to the expression of GAPDH. The cycling conditions were 95 °C for 15 s, followed by 40 cycles of 94 °C for 10 s, 60 °C for 30 s and 72 °C for 30 s. Primers used in this study were listed in Supplementary Table 1.

## Western blot analysis

Cells and tissues were lysed with RIPA buffer supplemented with protease inhibitor cocktail. Protein content was quantified using a BCA Protein Assay kit. All samples were reduced by boiling at 95 °C for 5 min in sodium dodecyl sulfate (SDS) sample buffer containing 5% beta-mercaptoethanol. Equal amounts of total protein were then loaded onto hand cast polyacrylamide gels. Proteins were resolved by sodium dodecyl sulfate-polyacrylamide gel electrophoresis (SDS-PAGE) and transferred to PVDF membranes. After blocking with 5% (wt/vol) skim milk in PBS with Tween 20 (0.1%) for 1 h, the membranes were incubated with the specific antibodies overnight at 4 °C, followed

by the appropriate HRP-conjugated second antibody 1 h at room temperature. Protein bands were detected using enhanced chemiluminescence reagent, and visualized on Tanon 4600 Imaging System (Tanon, China).

## Cell tube formation assay

96-well plates were coated with 50 µL/well Matrigel and FBS (1:1) which solidified at 37 °C for 30 min, according to the manufacturer's instructions. Cells were detached with trypsin, centrifuged and stained with 1 µM of Calcein-AM at 37 °C for 30 min. After washing with PBS twice, cells were resuspended in medium supplemented with 10% FBS. The cell suspensions were preincubated with GTN (44 µM), n-BANKs (60 µg/mL activated nanovesicles, 44 µM GTN) or PBS for 15 min and plated onto the surface of the polymerized Matrigel ($2 \times 10^4$ cells/well). After 3 h incubation at 37 °C, 5% $CO_2$, triplicate pictures were taken for each well using an EVOS FL Auto Cell Imaging System (Life Technologies). The angiogenesis parameters, including total tube length, numbers of branch points, master junctions and numbers of loops of tube-like structure in 9 fields per group were analyzed using the image analysis software ImageJ (NIH, USA).

## In vitro wound healing assay

Wound healing assay was performed on confluent SVEC4–10 in the present of 500 µM $CoCl_2$. Cells were seeded onto 12-well tissue culture plates and cultured in DMEM medium with 10% FBS until 90% of confluence. Three scratches were made across the quarter of each well using a sterile 200-µL pipet tip steadied with a ruler. Medium was changed twice after the scratching to remove detached and dead cells, and each scratch visually inspected. Scratches that were ragged, wider than the field of view or less than 1/3rd of the width of the field of view were excluded. Wound healing capacity was assessed by capturing phase-contrast images of the wounded area at the beginning and after 24 h under a EVOS FL Auto Cell Imaging System (Life Technologies). Images at 0 and 24 h for each experimental point were compared to quantify the migration rate of the cells after wounding. All experimental conditions were performed in triplicate.

## NO detection

For detection of intracellular NO generation, SVEC4–10 cells were first collected and washed with PBS for three times. Then, the cells were incubated with DAF-FM DA solution (dilution 1:1000) for 20 min at 37 °C. After washing the cells with PBS for three times, the cells were collected and resuspended in 500 µL PBS. The DAF fluorescence was analyzed using flow cytometry (CytoFLEX-S, Beckman Coulter, USA).

The released NO amount in culture medium was measured as the amounts of nitrites, determined on centrifuged medium by Griess reaction using the nitric oxide assay kit. For the NO detection in cells and tissues, cell and tissue lysis buffer for nitric oxide assay was used to lyse the SVEC4–10 cells and muscle tissues from mice. Then the mixture was centrifuged at 12,000 g for 5 min at 4 °C, and the NO was detected in the supernatant using nitric oxide assay kit.

## Real-time cell migration test

Pericyte migration was monitored in a CIM-16 well plate by using xCELLigence real time cell analyzer (RTCA) instrument (ACEA Biosciences, San Diego, CA, USA). Each well was composed of an upper and a lower chamber separated by a microporous membrane containing randomly distributed 8 µm-pores. A dimensionless parameter was measured as a cell-index which was used to evaluate the ionic environment at an electrode/solution interface and integrate information on cell numbers. As illustrated in Fig. 3G, 165 µL SVEC4–10 cell suspension ($5 \times 10^3$ cells) was loaded in the lower well of the CIM-16 plate. Following upper chamber attachment, the upper well was utilized with 45 µL prewarmed medium, and the plate left for 30 min at room temperature (RT) to pre-equilibrate. Pericytes were resuspended to

$2 \times 10^5$ cells/mL, and 100 µL cell suspension ($2 \times 10^4$ cells) was placed into each top well. The assembled plate was transferred to the RTCA-DP machine, and data were collected every 5 min over the course of 180 sweeps (15 h in total). This produces a signal of electrical impedance, which is reflected in the cell index as shown in the representative trace. Rising cell impedance therefore correlates to increasing numbers of migrated pericytes adhering to the lower chamber. Cells in lower chamber were collected for cell tube formation assay and collagen gel contraction assay.

## Collagen gel contraction assay

Gel contraction assay was performed using rat tail tendon collagen type I according to the manufacturer's instructions. Collagen type I was diluted to 2 mg/mL and adjusted to a pH of 7.4 with 1 M NaOH. Cells were mixed with type I collagen solution on ice. Cell-collagen mixture (500 µL) was plated into 48-well plates and collagen polymerization was initiated by incubating at 37 °C for 20 min. Using a sterilized spatula, the gels were gently transferred to 24-well plates. Gel contraction was initiated by releasing the gel from the side of the well, and changes in the collagen gel size were observed at 24 h.

## Immunofluorescence assay

Immunofluorescence was performed on paraffin-embedded muscle slices. After deparaffinization, slides were incubated with primary antibodies against NG2 (1:200), α-SMA (1:300) or eNOS (1:500) and subsequently with FITC conjugated anti-rabbit secondary antibodies (1:100), Cy3-conjugated anti-rabbit secondary antibodies (1:200) or FITC-conjugated anti-mouse secondary antibodies (1:100). DAPI (1 µg/mL) was used to visualize nuclei. Confocal immunofluorescence images of the tissues were captured on an Olympus Fluoview confocal microscope (FV3000, Olympus, Japan).

## Hindlimb ischemia model and perfusion imaging

Data from the Global Burden of Disease Study suggests that peripheral arterial disease cases in women outnumber those in men across all age groups[47]. Particularly, the prevalence of peripheral arterial disease in younger women in low-income or middle-income countries was 2.97 times more than that of men of the same age[48]. Critical limb ischemia (CLI) is a severe form of peripheral arterial disease. Thus, female mice were used to construct a CLI mouse model. Female BALB/c mice at 8–10 weeks of age were anesthetized via intraperitoneal injection with a combination of 100 mg/kg ketamine hydrochloride and 5 mg/kg xylazine. Depilatory cream was applied to the limbs and the area was sterilized by 70% ethanol applications. A 5-mm vertical skin incision was made lateral to the abdomen and superficial to the inguinal ligament. The inguinal fat pad was separated from the peritoneal lining to reveal the proximal femoral artery branching from the internal iliac artery. The femoral artery and vein were then separated from the membrane sheath and two ligatures were tied around both vessels approximately 2-mm apart. Vessels were transected between the ligatures and the skin incision was closed with two discontinuous sutures. Buprenorphine (0.5 mg/kg) was given twice within 24 h post-surgery. The drugs were intramuscularly injected into four sites of the gracilis muscle around the artery incision by using 1-mL syringes with 23-gauge needle.

Limb perfusion was monitored with a laser Doppler perfusion imaging system (Perimed, Inc., Ardmore, PA) at days 3, 7, and 14 post-surgery. Mice were placed on a warming pad during surgery and during laser Doppler image acquisition to maintain a constant body temperature of 37 °C. Perfusion was expressed as the ratio of the left (ischemic) to right (nonischemic) hindlimb. The right hindlimb served as an internal control for each mouse.

A square mask (with a size of $30 \times 30$ pixels) was drawn on the ankle region in the white light image as the selected representative ROI to monitor therapeutic efficacy. In addition, tissue damage in the

ischemic limb (limb salvage score) was graded as entirely recovery (grade 6), minor necrosis or nail loss (grade 5), partial toe amputation (grade 4), total toe amputation (grade 3), partial/total foot amputation (grade 2), or partial/total limb amputation (grade 1).

## Behavioral analysis

Mice were housed in a temperature of 20–26 °C and 40–70% humidity-controlled environment, under the 12 h/12 h dark/light cycle. All the behavioral tests were conducted during the light cycle phase in an enclosed behavior room. The behavioral experiments were performed for the mice in limb-salvage groups at 14 d after treatment.

For the grasping ring test, the mouse was grasped by the base of the tail and suspended above a blunt ring. Subsequently, the mouse was gently lowered towards the blunt ring and allowed to grasp the ring with its injured hindlimb. Then, the mouse was released and movies were used to record the grasping reflex, grip strength and latency to fall. Mice were allowed to rest for a minimum for 5 min before repeating the test.

For the gait analysis, the front and rear paws of the mice were coated with black and red nontoxic paints, respectively. Barriers were constructed to guide the mice to walk straight on the recording paper. The home cage was kept at the end of the recording paper to encourage completion of the test. Only trials in which the mouse made a continuous, direct path to its home cage were counted. The following gait parameters were then measured, (1) stride length, the distance between two successive rear paw prints on the left side; (2) stance length, the distance between the left and right rear paws; (3) intra-step distance, the vertical distance between the left and right rear paws; and (4) overlap length, the distance between the center of the front and rear paw prints on the left side.

## Serum biochemical analysis

Blood from the mice was drawn through the orbital venous plexus, and approximately a 1 mL portion of blood was collected from each mouse. Upon blood collection, full blood was let to clot for 15–30 min at room temperature. Blood samples were centrifuged at 3000 rpm for 20 min to harvest serum. Serum alanine aminotransferase (ALT), lactate dehydrogenase (LDH), creatinine (Cre) and blood urea nitrogen (BUN) levels were determined using an automatic biochemical analyzer (HITACHI, Japan).

## Histological analysis

On day 3 and day 14 of the ischemia model, the muscle samples from euthanized animals of each group were collected, then fixed with 10% formaldehyde overnight, and embedded in paraffin as we previously reported[49]. To evaluate fiber degeneration and apoptosis, muscle tissues were stained with hematoxylin and eosin (H&E). To estimate the degree of muscle fibrosis, Masson's trichrome staining was performed on sample tissues. All images were observed and captured under a EVOS FL Auto Cell Imaging System (Life Technologies).

## Statistical analysis

All statistical analysis was performed using Prism 7.0 software (GraphPad Software) by an unpaired Student's $t$-test, one-way or two-way ANOVA with Bonferroni multiple comparisons post-test. Statistical significance for survival curve was calculated by the log-rank test. Data were approximately normally distributed and variance was similar between the groups. Statistical significance is indicated as $*P < 0.05$, $**P < 0.01$, $***P < 0.001$, $****P < 0.0001$, and n.s. $> 0.05$.

## Reporting summary

Further information on research design is available in the Nature Portfolio Reporting Summary linked to this article.

## Data availability

The data supporting the findings from this study are available within the Article, Supplementary Information, or Source Data file. Source data are provided with this paper.

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

## Acknowledgements

We are indebted to Zhe Li for the help of molecular docking and Dai Gang for his guidance on laser Doppler measurements. This work was supported by National Natural Science Foundation of China (Project No. 82373812, 82360703), Guangdong Basic and Applied Basic Research Foundation (Project No. 2023A1515010011, 2021B1515120085), and Hainan Provincial Natural Science Foundation of China (823MS032). Dedicated to the 20th anniversary of School of Pharmaceutical Sciences, Sun Yat-sen University.

## Author contributions

L.G. and Q.Y. contributed equally to this work; L.G. and Q.Y. designed the experiments with the help from M.F.; L.G. and Q.Y. conducted most of the experiments; C.X. and C.L. performed the molecular docking; L.G., Q.Y., R.W., W.Z., N.Y., and Y.C. performed the data analysis and interpreted the results; L.G., R.P.C., Y.L. and M.F. wrote the manuscript and conceived the project.

## Competing interests

The authors declare no competing interests.
