## [Peer Review File · Nature Communications]

REVIEWER COMMENTS

Reviewer #1 (Remarks to the Author):

This manuscript reports the results of the development and evaluation of a nanoscale extracellular vesicle complexed with nitric oxide (NO)-boosting nanocage (n-BANK) for the treatment of critical limb ischemia. This study addresses an important issue concerning insufficiency in generating functional vascular structures in the ischemic limb. The authors produced a nanocage using an albumin-glyceryl trinitrate (GTN) complex, an agonist of eNOS. Extracellular vesicle (EV) from the bone marrow-derived mesenchymal stem cells (MSCs) was used as a carrier for NO-nanocage to form an n-BANK. The design of the n-BANK is clear and sound. In vitro results provided strong support for the characterization and functions of the n-BANK. Results from a mouse limb ischemia model demonstrated the effect of n-BANK treatment on enhancing angiogenesis and vascular formation, as well as recruiting pericytes to the newly formed vascular network. Importantly, results showed that n-BANK treatment improved limb preservation and restored the motor function of the limb. Overall, this manuscript is well written. Results and figures are clear and well laid out to support the conclusion. The n-BANK, developed in this study, is a novel nanotherapeutic agent that has the potential for further development as a new therapeutic agent.

To improve the clarity in the mechanism of n-BANK, it is important to determine the contributions of EVs as a carrier for the NO-nanocage or its biological activities to the mechanisms of the action and overall effect in the mouse model. Adding the No-nanocage only, and EV-only controls in several important in vitro and in vivo studies should increase the clarity of the results. Although one of the in vivo studies had the EV-only group and showed no therapeutic effect, it is unclear whether No-nanocage has a similar effect and if EVs affect the regulation of NO level or recruiting pericytes.

Additionally, Fig 1 (H) Illustration of pericyte recruitment via n-Bank is embedded in the results. Since there is no result supporting the illustration in Figure 1, it should be removed to an appropriate figure.

Reviewer #2 (Remarks to the Author):

The authors constructed nitric oxide (NO) donor carrying stem cell-derived extracellular vesicles (EVs)(n-BANKs). They report that n-BANKs taken up by endothelial cells increased pericyte recruitment and improved revascularization and functional recovery after critical limb ischemia. They propose that NO nanocages and multiple angiogenic factors shuttled by n-BANKs activated the endothelial nitric oxide

synthase (eNOS) in endothelial cells and enhanced eNOS-derived NO, which promoted vasodilation and nascent endothelial tube formation.

The strength of this manuscript is that its translational endpoint is very impressive. The authors conducted a series of elegant studies to develop NO carrying EVs and investigated their effects on the endothelium as well as revascularization.

The weaknesses of the manuscript are: 1) although it is very likely that NO released by EVs along with angiogenic factors play a role in the observed positive results, convincing mechanistic evidence has not been provided 2) presentation is sometimes not optimal, making assessment difficult although it is a generally a well written manuscript.

The role of NO and angiogenic factors in recovery have not been sufficiently differentiated. Inhibitors of eNOS (pharmacological or genetic) could have helped to disclose the magnitude of NO's contribution. This is important because it is possible that NO released may lead to unwanted effects in the ischemic environment where there is also oxygen radical formation, leading to toxic peroxynitrite formation. Do authors know what mechanisms ensure that NO is released in a physiologically optimal way but not in unregulated bursts? Janus-face of NO is well known.

The text written as a narrative of the experiments. Generally, no data are given regarding the number of experiments neither in the text, legend nor methods. Statistics like standard errors etc. can only be extrapolated from the figures. These should be provided in the text.

Line 195, using a co-culture method (Fig. S3B); there is no such figure

Line 284, overexpression of the eNOS gene. Please indicate the time after treatment. This eNOS mRNA increase time is unclear in the text as well as legend of Fig4B. Only Figs6A tells that this is 4 hours after treatment. However, here the order is different than what reads in the text; first phosphorylation and then transcriptional increase. Please clarify. Also indicate that these cells were subjected to hypoxia

330, reduced at 5 h after endothelial tube formation, suggesting regression occurred. Please clarify; do you mean 5h after hypoxia?

333, red fluorescent recruited pericytes; please indicate why pericytes are red

364, α -SMA antibody and eNOS antibody; their separate localization in endothelium and pericyte can only be differentiated with EM. Confocal cannot tell if they are colocalized in the same cell. Please clarify.

368, Local vasoconstriction (do you mean VASODILATION?) is thought to be accompanied by augmenting blood flow

Although the manuscript is well written, there are yet several grammatical errors throughout the text, some of which are listed below:

Line 151 collateral vessels for CLI treatment. To the end (to this end?...)

line 157 to that albumin could bound (bind?) vascular endothelial cells; similarly in line 190

line 188 ..showed the isolated activated EVs were (had?) cup-like concavity structure

197 It's likely due to (?)

288 ...cells were (?) a 3.13-fold increase

557, NO was withdrawn (???) from endothelial cells to facilitate pericyte recruitment.

The Discussion mainly reiterates the results with an emphasis on the treatment success. While this success is undeniably important, its relationship to NO is not entirely clear as the eNOS inhibition experiments were not performed. Again, contribution of the angiogenic factors provided by EVs has not been mechanistically clarified although EVs without nanocages were found not to be as effective. These issues should have been discussed as well.

Many undefined acronyms used in the Methods.

Figures are too busy; hence, the resolution of the panels is sometimes unsatisfactory. Legends are too brief, hence, making it difficult to understand for people who are not familiar with the methods.

Fig 1 legend 166, Cell viability of SVEC4-10 cells exposed to ischemic conditions; please indicate what ischemic conditions are.

Panel (C) is hard to assess pericyte drop out. Too small and low resolution. The same applies to D.

170, immunofluorescence images of muscle sections stained with α -SMA; please clarify ischemia duration. Resolution should be improved

D and N, please expand the legend.

Fig 2D Fluorescent analysis of n-BANKs (please indicate the fluorescent labels used)

G explain the storage time

Fig 3 please explain what the reader should see in A and C

Fig5 B, please give information about what to see for readers unfamiliar with the assay.

Fig 5C, Quantitative (do you mean quantification?)

Fig 2

Fluorescent

229 analysis of n-BANKs (indicate labels)

G explain storage time

Fig 3 please explain what the reader should see in A and C

380. Fig5 B give information about what to see for readers unfamiliar with the assay. Fig 5C (C)
Quantitative (quantification)

Reviewer #3 (Remarks to the Author):

In this study, the authors proposed and constructed n-BANK, a NO-boosted and activated nanovesicle regeneration kit with activated MSC-derived EVs and NO-nanocages to improve the vascularization and tissue regeneration in CLI. While the overall work is interesting major critiques need to address as commented below:

1. Authors highlighted the n-BANKs energize pericyte-endothelial interactions to create functional vascular networks, offering a new therapeutic platform for the treatment of CLI and other ischemia-related diseases. However, the key biological signalling pathways through which n-BANK achieves targeted vascular regeneration are not thoroughly explored in this study. They need to perform further experiments related to the signalling pathways involved in the current findings.

2. In this study, the authors used a strategy of "rescuing damaged endothelial cells and augmenting eNOS activity for pericyte recruitment". However, similar approaches have previously been reported. So, how does this study impact future outcomes in CLI therapy, aside from developing a unique n-BANK platform that aids in effective revascularization in a CLI?

3. Authors performed fluorescent labelling to evaluate the biodistribution of n-BANKs in ischemic tissues. Authors need to evaluate the long-term stability of the proposed platform in animals.

4. In Fig.1L, authors demonstrated that GTN trafficking into endothelial cells via albumin nanocages was more efficient than GTN alone, indicating the efficacy of albumin nanocages as a nanocarrier. Did the authors perform experiments other than molecular docking to confirm the high affinity of albumin nanocages? Please clarify what motivated the authors to choose albumin nanocages for the current study, although many other nanomaterials with multifunctionality that can serve as nanocarriers and target ischemic tissues have already been reported.

5. Albumin was one of the structural components of the n-BANK utilized in this work. According to the authors, the albumin part present on n-BANKs has the potential to attach to the albumin-binding glycoprotein (gp60), which is only expressed on the surface of vascular endothelial cells. Do the many functional groups in the albumin nanocage lead to any additional unwanted interactions, hence reducing the bioactivity of the protein nanocarrier? This point needs to be addressed properly.

Response to reviewers' comments:

Reviewer #1

Comments: This manuscript reports the results of the development and evaluation of a nanoscale extracellular vesicle complexed with nitric oxide (NO)-boosting nanocage (n-BANK) for the treatment of critical limb ischemia. This study addresses an important issue concerning insufficiency in generating functional vascular structures in the ischemic limb. The authors produced a nanocage using an albumin-glyceryl trinitrate (GTN) complex, an agonist of eNOS. Extracellular vesicle (EV) from the bone marrow-derived mesenchymal stem cells (MSCs) was used as a carrier for NO-nanocages to form an n-BANK. The design of the n-BANK is clear and sound. In vitro results provided strong support for the characterization and functions of the n-BANK. Results from a mouse limb ischemia model demonstrated the effect of n-BANK treatment on enhancing angiogenesis and vascular formation, as well as recruiting pericytes to the newly formed vascular network. Importantly, results showed that n-BANK treatment improved limb preservation and restored the motor function of the limb. Overall, this manuscript is well written. Results and figures are clear and well laid out to support the conclusion. The n-BANK, developed in this study, is a novel nanotherapeutic agent that has the potential for further development as a new therapeutic agent.

Thank you for the positive evaluation of our work. We have addressed all the points raised, as explained in detail here below.

Q1: To improve the clarity in the mechanism of n-BANK, it is important to determine the contributions of EVs as a carrier for the NO-nanocage or its biological activities to the mechanisms of the action and overall effect in the mouse model. Adding the No-nanocage only, and EV-only controls in several important in vitro and in vivo studies should increase the clarity of the results. Although one of the in vivo studies had the EV-only group and showed no therapeutic effect, it is unclear whether No-nanocage has a similar effect and if EVs affect the regulation of NO level or recruiting pericytes. Additionally, Fig 1 (H) Illustration of pericyte recruitment via n-Bank is embedded in

the results. Since there is no result supporting the illustration in Figure 1, it should be removed to an appropriate figure.

R1: We appreciate Reviewer#1's constructive suggestions. We have performed additional experiments to investigate the biological activities of EVs and NO-nanocages. As shown in Fig. R1A, the tube formation assay demonstrated that EVs could induce tubulogenesis and branching of endothelial cells, likely due to the EVs containing high levels of pro-angiogenic growth factors such as VEGF (Fig. 1D-F). However, EVs did not promote NO production in endothelial cells within 24 h (Fig. R1B). On the other hand, NO-nanocages could efficiently promote NO synthesis, but had no impact on the endothelial cell migration that is the first step in angiogenesis occurrence (Fig. R1C). Moreover, therapeutic effects of NO-nanocages in a mouse model of hind-limb ischemia showed that NO-nanocages did not induce new blood-vessel formation around the ligation sites of femoral artery and saphenous artery at day 3 following severe ischemic injury (Fig. R1D). In contrast, notable newly formed vessels could be detected in n-BANK treated mice (Fig. 7A). Additionally, we have deleted the Fig. 1 (H) from Figure 1. Based on the new data, we have rewritten the section (on page 13, lines 261-262; page 16, line 305) and included this data as new Supplementary Figs. 4, 7B and 9C in the revised manuscript.

Figure R1. Effects of EVs and NO-nanocages on the regulation of NO levels and new blood-vessel formation. (A) Tube formation was monitored periodically with images depicting endothelial tube formation 4 h after seeding. Scale bar, 1000 μm . (B) Release of NO in the cell supernatants of SVEC4-10 cells treated with EVs was quantified using the Griess reagent ($n = 6$). (C) Representative photomicrographs and quantification of a scratch-cell motility assay of SVEC4-10 cells under ischemic conditions for the indicated times ($n = 3$). Scale bar, 400 μm . (D) Photographs showing new vessel formation across ligation sites of the ischemic hindlimbs at day 3 after NO-nanocage treatment. All data are expressed as mean \pm SD, n.s. is $P > 0.05$ by Student's t-test.

Reviewer#2

Comments: The authors constructed nitric oxide (NO) donor carrying stem cell-derived extracellular vesicles (EVs) (n-BANKs). They report that n-BANKs taken up by endothelial cells increased pericyte recruitment and improved revascularization and functional recovery after critical limb ischemia. They propose that NO nanocages and multiple angiogenic factors shuttled by n-BANKs activated the endothelial nitric oxide synthase (eNOS) in endothelial cells and enhanced eNOS-derived NO, which promoted vasodilation and nascent endothelial tube formation. The strength of this manuscript is that its translational endpoint is very impressive. The authors conducted a series of elegant studies to develop NO carrying EVs and investigated their effects on the endothelium as well as revascularization. The weaknesses of the manuscript are: 1) although it is very likely that NO released by EVs along with angiogenic factors play a role in the observed positive results, convincing mechanistic evidence has not been provided 2) presentation is sometimes not optimal, making assessment difficult although it is a generally a well written manuscript.

Thank you for your careful reading of the manuscript and thoughtful suggestions that have helped us improve the manuscript. Please find our point-by-point responses to the comments below.

Q2: The role of NO and angiogenic factors in recovery have not been sufficiently differentiated. Inhibitors of eNOS (pharmacological or genetic) could have helped to disclose the magnitude of NO's contribution. This is important because it is possible that NO released may lead to unwanted effects in the ischemic environment where there is also oxygen radical formation, leading to toxic peroxynitrite formation. Do authors know what mechanisms ensure that NO is released in a physiologically optimal way but not in unregulated bursts? Janus-face of NO is well known.

R2: We thank Reviewer#2 very much for the valuable suggestions. The eNOS inhibition assay by *N*^G-nitro-L-arginine methyl ester (L-NAME) was performed to investigate the magnitude of NO's contribution. A fluorescence probe, 4-amino-5-methylamino-2',7'-difluorofluorescein diacetate (DAF-FM DA), was used to detect

intracellular NO in endothelial cells and the results showed that the NO-producing cells displayed a sharp decrease from 83.80% to 43.46% after treatment with the eNOS inhibitor L-NAME (Fig. R2A and R2B). Next, we conducted the Boyden chamber assay to investigate the migration response of pericytes to L-NAME treated endothelial cells. The results demonstrated that inhibition of NO synthesis in endothelial cells could markedly reduce recruitment of pericytes to endothelial cells (Fig. R2C and R2D). Collectively, NO produced by endothelial cells played a crucial role in energizing pericyte-endothelial interactions.

We totally agree with the Reviewer#2's comment about Janus-face of NO. However, NO synthesis is significantly impaired in peripheral arterial diseases including critical limb ischemia (Circulation, 2007, 116, 188-195; Nat Rev Drug Discov., 2015, 14, 623-641). Thus, in the present study, we constructed n-BANKs to repair NO synthesis by regulation of endothelial nitric oxide synthase (eNOS) expression and phosphorylation. The eNOS-derived NO can serve in its capacity as a vasodilator to provide blood flow and nutrients to the ischemic tissues. Furthermore, eNOS is a fundamental mediator of vascular function maintaining endothelial homeostasis, which could control NO release in a physiologically optimal way but not in unregulated bursts. (PNAS, 2005, 102, 10999-11004; Cell, 2022, 185, 2853-2878).

Figure R2. eNOS-derived NO promoted pericyte recruitment. (A) Flow cytometry analysis of NO levels in SVEC4-10 cells. (B) Quantification of NO-producing cells by flow cytometry (n = 3). (C) Cell migration assay by an 8.0 μm transwell revealed pericyte recruitment by n-BANKs and the eNOS inhibitor L-NAME. The non-migrating cells in the upper chambers were stained by crystal violet, and the migrating cells in the lower chambers were stained with calcein acetoxymethyl ester (calcein AM). (D) Quantification of pericytes in the upper chambers and the lower chambers (n = 3). All data are expressed as mean ± SD, ** is $P < 0.01$, *** is $P < 0.001$ and **** is $P < 0.0001$ by Student's t-test.

Q3: The text written as a narrative of the experiments. Generally, no data are given regarding the number of experiments neither in the text, legend nor methods. Statistics like standard errors etc. can only be extrapolated from the figures. These should be provided in the text.

R3: Thank reviewer#2 for the suggestion. We have provided the numerical values of

the critically measured parameters, the number of experiments and the p-value calculated from statistical tests in the revised manuscript.

Q4: Line 195, using a co-culture method (Fig. S3B); there is no such figure
Line 284, overexpression of the eNOS gene. Please indicate the time after treatment. This eNOS mRNA increase time is unclear in the text as well as legend of Fig4B. Only Figs6A tells that this is 4 hours after treatment. However, here the order is different than what reads in the text; first phosphorylation and then transcriptional increase. Please clarify. Also indicate that these cells were subjected to hypoxia
330, reduced at 5 h after endothelial tube formation, suggesting regression occurred. Please clarify; do you mean 5h after hypoxia?

333, red fluorescent recruited pericytes; please indicate why pericytes are red

R4: Line 195 (line 202 in revised manuscript), we have replaced “Fig. S3B” with “Supplementary Fig. 5B”.

Line 284, the overexpression of the eNOS gene occurred at as early as 4 h after treatment, which was included in the revised manuscript (on page 17, line 330). Moreover, the eNOS transcriptional increase accompanied with the protein phosphorylation, and there was no obvious relevance between them. The description has been modified in the manuscript (on page 15, lines 299-301). We have clarified that these cells were subjected to hypoxia in the legend of Figure 4 and Figure S9.

Line 330, “5 h” indicated the 5 h after endothelial tube formation, which was 9 h after treatment. We have clarified the time in the revised manuscript (on page 21, lines 403-404).

Line 333 (line 405 in revised manuscript), pericytes were stained with DiI and showed red fluorescent after excitation. We have included it in the legend of Figure 5.

Q5: 364, α -SMA antibody and eNOS antibody; their separate localization in endothelium and pericyte can only be differentiated with EM. Confocal cannot tell if they are colocalized in the same cell. Please clarify.

368, Local vasoconstriction (do you mean VASODILATON?) is thought to be

accompanied by augmenting blood flow.

R5: We totally agree with the comment regarding the separated localization of α -SMA antibody and eNOS antibody separate localization in endothelium and pericytes, respectively. As shown in Fig. 6C, ischemic muscle sections were stained with α -SMA antibody and eNOS antibody. The α -SMA antibody was used to identify pericytes while the eNOS antibody labeled endothelial cells in the ischemic muscle sections. The immunofluorescence images showed that n-BANKs induced a significant increase in eNOS expression in ischemic muscle tissues, and the mean internal diameter of α -SMA-positive blood vessels in n-BANK treated groups was significantly larger than that of the untreated control (Fig. 6C). Additionally, we have replaced the “vasoconstriction” with “vasodilation” (line 388).

Q6: Although the manuscript is well written, there are yet several grammatic errors throughout the text, some of which are listed below:

Line 151 collateral vessels for CLI treatment. To the end (to this end?...)

line 157 to that albumin could bound (bind?) vascular endothelial cells; similarly in line 190

line 188 ..showed the isolated activated EVs were (had?) cup-like concavity structure

197 It’s likely due to (?)

288 ...cells were (?) a 3.13-fold increase

557, NO was withdrawn (???) from endothelial cells to facilitate pericyte recruitment.

R6: We have changed “to the end” to “to this end” (line 152), and corrected “bound” to “bind” (lines 158 and 198). Besides, we changed “EVs were cup-like concavity structure” to “EVs had cup-like concavity structure” (line 195), “It’s likely due to” to “The loss of fluorescence signal was likely caused by” (lines 204-205), “cells were a 3.13-fold increase” to “cells had a 3.13-fold increase” (line 303), and “was withdrawn” to “released” (line 587). We also have carefully revised the whole manuscript and tried to avoid any grammar mistakes.

Q7: The Discussion mainly reiterates the results with an emphasis on the treatment

success. While this success is undeniably important, its relationship to NO is not entirely clear as the eNOS inhibition experiments were not performed. Again, contribution of the angiogenic factors provided by EVs has not been mechanistically clarified although EVs without nanocages were found not to be as effective. These issues should have been discussed as well.

R7: Thanks Reviewer#2 for the insightful suggestion. We have conducted additional experiments to investigate the magnitude of NO's contribution to therapeutic efficacy by using an eNOS inhibitor L-NAME. The results showed that the NO-producing cells displayed a sharp decrease from 83.80% to 43.46% after treatment with an eNOS inhibitor L-NAME (Fig. R2A and R2B). Next, we have performed the Boyden chamber assay to investigate the migration response of pericytes to L-NAME treated endothelial cells. The results demonstrated that pericyte migration was significantly inhibited after L-NAME acting on endothelial cells, suggesting that inhibition of NO synthesis in endothelial cells could markedly reduce recruited pericyte population (Fig. R2C and R2D). We have also performed the scratch assay and endothelial cell tube formation assay to assess the contributions of the angiogenic factors provided by EVs to therapeutic efficacy *in vitro*. As shown in Fig. R3, EV-activated endothelial cells migrated significantly farther and formed much more endothelial tubes than the untreated cells. However, EVs were not effective against ischemic injury *in vivo*, most likely due to that EVs without NO-nanocages could not recruit pericytes effectively to build well-functioning mature blood vessels. We have included the explanation (on page 23, line 450-452) and this data as new Supplementary Fig. 4 in the revised manuscript.

Figure R3. EVs promoted endothelial tube formation. (A) Representative photomicrographs of a scratch-cell motility assay of SVEC4-10 cells treated with EVs under ischemic conditions. Scale bar, 400 μm . (B) Tube formation was monitored periodically with images depicting endothelial tube formation 4 hours after seeding. Scale bar, 1000 μm . (C) Quantification of vascular progression (number of loops, vascular length, number of branching points and number of master junctions) at 4 h ($n = 15$). Data are mean \pm SD, ** is $P < 0.01$, *** is $P < 0.001$, **** is $P < 0.0001$ by Student's t-test.

Q8: Many undefined acronyms used in the Methods.

R8: We have defined the acronyms at first mention in the Methods. In addition, we have examined all acronyms in the revised manuscript that have been defined upon their first appearance.

Q9: Figures are too busy; hence, the resolution of the panels is sometimes unsatisfactory. Legends are too brief, hence, making it difficult to understand for people who are not familiar with the methods.

Fig 1 legend 166, Cell viability of SVEC4-10 cells exposed to ischemic conditions; please indicate what ischemic conditions are. Panel (C) is hard to assess pericyte drop out. Too small and low resolution. The same applies to D.170, immunofluorescence images of muscle sections stained with α -SMA; please clarify ischemia duration. Resolution should be improved. D and N, please expand the legend.

Fig 2D Fluorescent analysis of n-BANKs (please indicate the fluorescent labels used)
G explain the storage time

Fig 3 please explain what the reader should see in A and C

Fig5 B, please give information about what to see for readers unfamiliar with the assay.

Fig 5C, Quantitative (do you mean quantification?)

R9: We have incorporated Reviewer#2's suggestion throughout the manuscript. We have increased the resolution of the panels and revised the figure legends to be succinct but comprehensive for making our figures easy to understand.

In Fig. 1, SVEC4-10 cells were incubated in the presence of 500 μ M cobalt chloride (CoCl_2), a hypoxia-mimetic agent, to mimic hypoxic/ischemic conditions (Science, 1987, 242, 1412-1415; Sci Transl Med., 2020, 12, eaay0271). We have included stimulation of the mimic ischemic condition in the legend of Fig. 1A. Additionally, the duration of ischemia was 3 days in Fig. 1C (Fig. 1B in revised manuscript) and we have clarified it in the legend. We have increased the resolution of the Fig. 1 C and D (Fig. 1B and C in revised manuscript), and expanded the legend of the Fig. 1 D and N (Fig. 1C and L in revised manuscript).

In Fig. 2, n-BANKs were labeled with DiI dye in Fig. 2D. As shown in Fig. 2G, n-BANKs were shown to be stable for a minimum of 3 days if refrigerated at 4°C. We have included explanation of the storage time in the legend of Fig. 2G.

Endothelial cell migration and tube formation play crucial roles in angiogenesis. As shown in Fig. 3A, the scratch assay was performed to assess whether n-BANKs could enhance endothelial cell migration. The results indicated that endothelial cells treated with n-BANKs migrated significantly farther than cells exposed to either the GTN control or medium alone. Next, the ability of n-BANKs to induce endothelial tube formation was evaluated by *in vitro* tube formation assay, as shown in Fig. 3C. The

results showed that n-BANKs induced marked tubulogenesis, with the formation of tubules assembled by elongation and joining of endothelial cells. Taken together, these data suggested that n-BANKs could promote endothelial cell migration and endothelial tube formation.

Mature microvascular structure is characterized by pericytes wrap around the endothelial tubes. In Fig. 5B, we performed the collagen gel contraction assay to investigate whether n-BANK energizing pericyte-endothelial cell interactions was favorable to mature microvessel formation. Endothelial cells and the recruited pericytes were embedded into a collagen matrix and the contractility was monitored. The results showed that n-BANK treated group displayed a greater increase in contractile activity than the GTN treated and untreated control groups. In addition, we have corrected “Quantitative” to “Quantification” in Fig. 5C. The revision has been made and highlighted in yellow color in the revised manuscript.

Reviewer#3

Comments: In this study, the authors proposed and constructed n-BANK, a NO-boosted and activated nanovesicle regeneration kit with activated MSC-derived EVs and NO-nanocages to improve the vascularization and tissue regeneration in CLI. While the overall work is interesting, major critiques need to address as commented below:

We thank Reviewer#3 for the comments and suggestions, and for considering our work interesting. Below we have addressed the concerns point by point.

Q10: Authors highlighted the n-BANKs energize pericyte-endothelial interactions to create functional vascular networks, offering a new therapeutic platform for the treatment of CLI and other ischemia-related diseases. However, the key biological signalling pathways through which n-BANK achieves targeted vascular regeneration are not thoroughly explored in this study. They need to perform further experiments related to the signalling pathways involved in the current findings.

R10: We appreciate Reviewer#3's suggestion. We performed additional experiments to investigate the eNOS-NO signaling axis functions induced by n-BANKs to promote pericyte-endothelial interactions for mature vascular regeneration. As shown in Fig. 4, n-BANKs could activate eNOS of endothelial cells to produce NO, which played a crucial role in effective pericyte recruitment to stabilize and function nascent endothelial tubes. Furthermore, when NO synthesis in endothelial cells was inhibited by an eNOS inhibitor L-NAME (Fig. R4A), endothelial cell-driven pericyte migration was inefficient by using the chamber Boyden assay, suggesting that pharmacological inhibition of NO synthesis in endothelial cells could markedly reduce recruitment of pericytes (Fig. R4B-D). Collectively, eNOS-NO signaling axis played a crucial role in energize pericyte-endothelial interactions. Thanks again for the reviewer's suggestion, we hope the reviewer allows us to comprehensively address how the eNOS-NO signaling pathways through which n-BANK achieves targeted vascular regeneration in future work. Additionally, we have included this issue in the limitations of the study (on page 32, lines 616-617).

Figure R4. eNOS-NO signaling axis functions induced by n-BANKs for pericyte-endothelial interactions. (A) Flow cytometry analysis of NO levels in SVEC4-10 cells (n = 3). (B) Schematic illustration of cell migration assays by an 8.0 μm transwell. (C) The non-migrating cells in the upper chambers were stained by crystal violet and (D) the migrating cells in the lower chambers were stained with calcein acetoxymethyl ester (calcein AM) (n = 3). All data are expressed as mean \pm SD, ** is $P < 0.01$, *** is $P < 0.001$ and **** is $P < 0.0001$ by Student's t-test.

Q11: In this study, the authors used a strategy of "rescuing damaged endothelial cells and augmenting eNOS activity for pericyte recruitment". However, similar approaches have previously been reported. So, how does this study impact future outcomes in CLI therapy, aside from developing a unique n-BANK platform that aids in effective revascularization in a CLI?

R11: Thank Reviewer#3 for the insightful comments. Aside from developing the unique n-BANK platform that aids in effective revascularization in a CLI, we hope that our work could provide a potential new therapeutic strategy and spark a discussion of

CLI therapy to reduce high amputation rates of CLI. Given diabetes, hypertension and dyslipidemia raise the risk of developing CLI, it is likely to be an increasingly important issue. Patients with CLI have a major amputation rate as high as 40% and amputation is associated with poor survival outcomes (Nat. Rev. Cardiol. 2013, 10, 387-396; Nature 548, 2017, S41). In CLI patients, a foot or leg may be removed by surgical amputation if blood flow cannot be returned to the ischemic foot or leg. The n-BANK facilitates maturity of newly formed vessels to build functional vascular networks *via* energizing pericyte-endothelial interactions, which differs from either stem cell therapy or eNOS gene therapy. Furthermore, with the development of early detection of ischemic diseases, the therapeutic strategy of effective revascularization would play a more important role in the treatment of peripheral artery disease, the early stages of CLI, without the need for amputation.

Q12: Authors performed fluorescent labelling to evaluate the biodistribution of n-BANKs in ischemic tissues. Authors need to evaluate the long-term stability of the proposed platform in animals.

R12: To investigate the long-term stability of n-BANKs *in vivo*, Cy5-labeled n-BANKs were administered (*i.m.*) into BALB/c mice with severe ischemic hindlimbs and monitored using a real-time *in vivo* imaging system (NightOWL II LB983). As shown in Fig. R5, fluorescence signal intensity was evaluated as a function of n-BANK concentration. The n-BANKs showed a strong fluorescence after *i.m.* injection, and fluorescence signals decreased gradually to the baseline at 66 h. The results demonstrated in mice that n-BANKs stayed at the ischemic hindlimb for more than 2 days. We have included the explanation (on page 11, lines 220-221) and the new data (Supplementary Fig. 6) in the revised manuscript.

Figure R5. Live imaging and quantitation of fluorescence intensity of FITC-labeled n-BANKs after *i.m.* injection at the indicated times (n = 3).

Q13: In Fig. 1L, authors demonstrated that GTN trafficking into endothelial cells via albumin nanocages was more efficient than GTN alone, indicating the efficacy of albumin nanocages as a nanocarrier. Did the authors perform experiments other than molecular docking to confirm the high affinity of albumin nanocages? Please clarify what motivated the authors to choose albumin nanocages for the current study, although many other nanomaterials with multifunctionality that can serve as nanocarriers and target ischemic tissues have already been reported.

R13: We have performed surface plasmon resonance to further confirm the affinity of albumin and GTN. As shown in Fig. R6, increasing concentrations of GTN showed proportional increases in binding to albumin, suggesting GTN had a binding affinity to albumin. Compared with other multifunctional nanomaterials, albumin has excellent biocompatibility and its successful clinical applications. Importantly, albumin has a

high affinity to both GTN and endothelial cells, which is favorable to encasing GTN in albumin nanocages and delivering therapeutic agents to endothelial cells. Therefore, we chose albumin nanocages for the current study. We have included this data as new Supplementary Fig. 5A in the revised manuscript.

Figure R6. Surface plasmon resonance determination of GTN binding affinity to albumin.

Q14: Albumin was one of the structural components of the n-BANK utilized in this work. According to the authors, the albumin part present on n-BANKs has the potential to attach to the albumin-binding glycoprotein (gp60), which is only expressed on the surface of vascular endothelial cells. Do the many functional groups in the albumin nanocage lead to any additional unwanted interactions, hence reducing the bioactivity of the protein nanocarrier? This point needs to be addressed properly.

R14: We appreciate Reviewer#3 for the suggestion. Indeed, albumin is able to bind endogenous molecules, such as long-chain fatty acids, steroids, and L-tryptophan. However, albumin is one of the most abundant proteins in the body. Around 40% of albumin is present in the plasma and 60% in the extracellular space (Hepatology, 2013, 58, 1836-1846). The abundant endogenous albumin could limit unwanted interactions between albumin nanocages of n-BANK and endogenous molecules. Our results from a mouse limb ischemia model demonstrated that n-BANK could work well to markedly improve limb preservation and restore the motor function of the limb. We have included the explanation (on page 11, lines 227-229) in the revised manuscript.

REVIEWER COMMENTS

Reviewer #1 (Remarks to the Author):

The revised manuscript has addressed my previous comments very well. It is acceptable for publication.

Reviewer #3 (Remarks to the Author):

The revised version addressed well the comments.

Reviewer #4 (Remarks to the Author):

I thank the authors as they responded point-by-point to the comments of the reviewer and improved the manuscript. However, there are minor points that will be better if clarified.

The comment 5 is not satisfactorily elucidated. The authors indicate that the α -SMA antibody was used to identify pericytes. It is known that pericytes are located at capillaries. However, the α -SMA-positive blood vessels demonstrated in Figure 6 A and C are not capillaries, hence probably the cells are not pericytes when the vessel diameters are checked from the scale bar. The authors should use other approved pericyte markers if they specifically want to identify the pericytes.

It will be better to put the explanation written in the rebuttal letter for the comment Fig 3 -that explain what the reader should see in A and C- to the figure legend of the manuscript.

The fluorescence images should be clarified. The authors reveal that EVs were prelabeled by Dil. Then, it is also written that pericytes were labeled with Dil (red). Hence the fluorescence imaging revealed in Fig 4 I is still not clear.

It will be better if the sentence 'However, GTN and NO-nanocage had no impact on the endothelial cell migration (Supplementary Fig. 7B)' is written before 'The promigratory effects of n-BANKs were confirmed under normoxia (Supplementary Fig. 8)'

Response to reviewers' comments:

Reviewer #1

Comments: The revised manuscript has addressed my previous comments very well. It is acceptable for publication.

R: We thank Reviewer#1 for your previous comments that helped us improve the manuscript.

Reviewer #3

Comments: The revised version addressed well the comments.

R: We thank Reviewer#3 for your previous comments that helped us improve the manuscript.

Reviewer #4

Comments: I thank the authors as they responded point-by-point to the comments of the reviewer and improved the manuscript. However, there are minor points that will be better if clarified.

R: We sincerely appreciate Reviewer#4's constructive comments and suggestions, which helped us in improving the quality of the manuscript. Our point-by-point responses to the comments are described as below.

Q1: The comment 5 is not satisfactorily elucidated. The authors indicate that the α -SMA antibody was used to identify pericytes. It is known that pericytes are located at capillaries. However, the α -SMA-positive blood vessels demonstrated in Figure 6 A and C are not capillaries, hence probably the cells are not pericytes when the vessel diameters are checked from the scale bar. The authors should use other approved pericyte markers if they specifically want to identify the pericytes.

R: We thank Reviewer#4 for the suggestion. Evidence shows that the NG2 proteoglycan is expressed by microvascular pericytes in newly formed blood vessels (Nat Immunol., 2013, 14, 41-51; Circulation, 2020, 142: 688-704). Thus, we used the NG2 proteoglycan as a marker to investigate the recruitment of pericytes to blood

vessels. Fig. R1 shows the immunostaining of NG2-positive cells in sections from ischemic limbs. We observed a very low number of NG2-positive pericytes in either untreated or GTN treated ischemic limbs, indicating the poor pericyte recruitment. However, n-BANK treatment significantly increased the number of NG2-positive pericytes in ischemic limbs. We have revised and highlighted this part of the revised manuscript in yellow (on page 19, lines 373-378 and page 22, lines 415-417).

Figure R1. Representative images and quantification showing NG2 expression and pericyte coverage in ischemic muscle sections. Scale bar, 100 μ m. Data are mean \pm SD, * is $P < 0.05$, ** is $P < 0.01$ by one-way ANOVA test.

Q2: It will be better to put the explanation written in the rebuttal letter for the comment Fig 3 -that explain what the reader should see in A and C- to the figure legend of the manuscript.

R: We appreciate Reviewer#4 for the suggestion. We have included the explanation in the revised manuscript (on page 15, lines 282-283 and lines 285-287).

Q3: The fluorescence images should be clarified. The authors reveal that EVs were pre-labeled by DiI. Then, it is also written that pericytes were labeled with DiI (red). Hence the fluorescence imaging revealed in Fig 4 I is still not clear.

R: In Fig 4I, EVs were not fluorescently labeled while pericytes were labeled with calcein acetoxymethyl ester (Calcein-AM). We have added a note on this in the figure legend to avoid such confusion (on page 18, line 346).

Q4: It will be better if the sentence ‘However, GTN and NO-nanocage had no impact on the endothelial cell migration (Supplementary Fig. 7B)’ is written before ‘The promigratory effects of n-BANKs were confirmed under normoxia (Supplementary Fig. 8)’

R: We thank Reviewer#4 for the suggestion. We have rewritten the sentence (on page 13, lines 261-263).

REVIEWERS' COMMENTS

Reviewer #4 (Remarks to the Author):

The revised version 2 addressed the comments well, and cleared my concerns.

Response to reviewers' comments:

Reviewer #4

Comments: The revised version 2 addressed the comments well, and cleared my concerns.

R: We thank Reviewer#4 for your previous comments that helped us improve the manuscript.